# To Stay or Not to Stay in the Pre-train Basin: Insights on Ensembling in Transfer Learning

**Ildus Sadrtdinov[1]***, **Dmitrii Pozdeev[1,2]***, **Dmitry Vetrov[3], Ekaterina Lobacheva[4]**

[1]HSE University  [2]Technical University of Munich (TUM)
[3]Constructor University, Bremen  [4]Independent researcher
isadrtdinov@hse.ru, dmitrii.pozdeev@tum.de
dvetrov@constructor.university, lobacheva.tjulja@gmail.com

## Abstract

Transfer learning and ensembling are two popular techniques for improving the performance and robustness of neural networks. Due to the high cost of pre-training, ensembles of models fine-tuned from a single pre-trained checkpoint are often used in practice. Such models end up in the same basin of the loss landscape, which we call the pre-train basin, and thus have limited diversity. In this work, we show that ensembles trained from a single pre-trained checkpoint may be improved by better exploring the pre-train basin, however, leaving the basin results in losing the benefits of transfer learning and in degradation of the ensemble quality. Based on the analysis of existing exploration methods, we propose a more effective modification of the Snapshot Ensembles (SSE) for transfer learning setup, StarSSE, which results in stronger ensembles and uniform model soups.

## 1 Introduction

Ensembling neural networks by averaging their predictions is one of the standard ways to improve model quality. This technique is known to improve not only the task-specific quality metric, e. g., accuracy, but also the model calibration (Lakshminarayanan et al., 2017) and robustness to noise (Gustafsson et al., 2020), domain shift (Ovadia et al., 2019), and adversarial attacks (Strauss et al., 2017). In practice, the importance of the solution robustness often goes hand in hand with the scarcity of data and computational resources for training. A popular way to obtain a high-quality solution for a task with limited data using less computation is transfer learning, which initializes the training process on the target task with the model checkpoint pre-trained on a large source task.

In this paper, we investigate the effectiveness of neural network ensembling methods in transfer learning setup since combining the two techniques is a non-trivial problem. The diversity of networks is crucial for obtaining substantial quality gain with ensembling. Therefore, in classic deep ensembles (DE by Lakshminarayanan et al. (2017)), networks are trained independently from different random initializations. However, in the transfer learning setup, networks are not trained from scratch but are fine-tuned from pre-trained checkpoints. Fine-tuning networks from different independently pre-trained checkpoints results in a diverse set of models and a high-quality final ensemble but requires a huge amount of computation due to the large cost of the pre-training process. Fine-tuning networks from the same pre-trained checkpoint is much more resource-effective; however, the resulting set of networks are usually located in the same basin of the loss landscape (Neyshabur et al., 2020), are less diverse, and their ensemble has lower quality (Mustafa et al., 2020). So, the main question of our paper is the following: can we improve the quality of the ensemble of networks obtained by fine-tuning from the same pre-trained checkpoint by better exploring the loss landscape of the target task?

---

*First two authors contributed equally.

37th Conference on Neural Information Processing Systems (NeurIPS 2023).

We focus on the image classification target tasks and consider two types of pre-training: supervised (Kornblith et al., 2019) and self-supervised (Grill et al., 2020). First of all, in Section 5.1, we confirm that in this setup, there is indeed a substantial quality gap between ensembles of models fine-tuned from the same and different pre-trained checkpoints. Thus, methods aiming at closing this gap are worth researching. We also show that models fine-tuned from the same pre-trained checkpoint lay in the same basin of the loss landscape, meaning there is no significant high-loss barrier on the linear interpolation between them. We refer to this basin as the pre-train basin throughout the paper.

To understand whether the ensemble quality can be improved by better exploring the pre-train basin or the close vicinity outside of it, we take inspiration from the literature on efficient ensembling methods in regular non-transfer learning setup. We focus our study on cyclical learning rate methods, such as Snapshot Ensembling (SSE by Huang et al. (2017)), because they can either explore one basin of the loss landscape or go outside of the basin depending on the chosen hyperparameters. In Section 5.2, we show that with accurate hyperparameter tuning, these methods can explore the pre-train basin and produce ensembles of similar or even higher quality than the independent fine-tuning of several neural networks. However, they are not able to leave the pre-train basin without losing the advantages of transfer learning, followed by significant degradation in the quality of individual models and the whole ensemble. Thus, we conclude that the gap between ensembles of networks fine-tuned from the same and from different independently pre-trained checkpoints can not be closed by exploring the loss landscape of the target task using existing methods.

Based on the analysis of the behavior of the standard sequential variant of SSE in Section 5.3, we propose its parallel variant StarSSE, which is more suitable for exploring the pre-train basin in the transfer learning setup (see Section 5.4). StarSSE fine-tunes the first model from the pre-trained checkpoint in a usual way and then trains all the following models, independently initializing them with the first fine-tuned one. In contrast to the SSE, StarSSE does not move further from the pre-trained checkpoint with each training cycle; therefore, it does not suffer from the progressive degradation of individual model quality. Even though StarSSE is technically very similar to the independent fine-tuning of all the models from the pre-trained checkpoint, it has an important conceptual difference: it separates 1) model fine-tuning from the pre-train checkpoint to the area with high-quality solutions for the target task and 2) exploration of this area for obtaining diverse models. As a result, StarSSE can collect a more diverse set of networks without sacrificing the quality of individual models.

Finally, we analyze different aspects of SSE and StarSSE that can be important for their practical application. Firstly, in Section 6, we show that the set of models, collected by SSE or StarSSE in the pre-train basin, can not only constitute an effective ensemble but also result in a strong uniform model soup (Wortsman et al., 2022; Rame et al., 2022), i. e., a model obtained by weight-averaging all models in the set. Then, in Section 7, we study the robustness of the resulting ensembles and model soups to the test data distribution shifts. Lastly, in Section 8, we confirm that SSE and StarSSE are effective in more practical large-scale experiments with CLIP model (Radford et al., 2021) on the ImageNet classification target task (Russakovsky et al., 2015). In all additional experiments, StarSSE also outperforms the standard SSE.

Our code is available at `https://github.com/isadrtdinov/ens-for-transfer`.

## 2 Ensembling methods in regular training

Deep ensemble (DE) is the most common method of neural network ensembling in a regular non-transfer learning setup. A DE consists of several networks of the same architecture trained separately from different random initializations using the same training methods and hyperparameters. Training from different initializations makes neural networks diverse and results in a high-quality ensemble. However, separate training of all the networks requires a lot of resources. From the loss landscape perspective, DE can be called a global method: due to diverse initializations, networks in such an ensemble are independent and come from different basins of low loss (Fort et al., 2019).

To reduce training costs, many methods propose to train the networks jointly or consequentially. Most of these methods can be assigned to one of two groups based on the degree of coverage of different loss landscape basins: local and semi-local. Local methods (SWAG by Maddox et al. (2019), SPRO by Benton et al. (2021), FGE by Garipov et al. (2018), etc.) train the first network and then efficiently sample other ensemble members from the basin of low loss around it. Semi-local methods (SSE

by Huang et al. (2017), cSGLD by Zhang et al. (2020), etc.), instead of focusing on one basin, try to explore a larger portion of the loss landscape by sequentially training networks using a cyclical learning rate schedule. At the beginning of each cycle, the learning rate is increased to help the network leave the current low-loss basin, and at the end of the cycle, it is decreased to obtain a high-quality checkpoint in the next basin to add to the ensemble. Because of the knowledge transfer between the consecutive models, such methods can use training cycles shorter than usual model training, which decreases the overall ensemble training cost. In terms of network diversity and the final ensemble quality, semi-local methods work better than local ones, given the same number of networks in an ensemble. However, methods from both of these groups are inferior to the global ones (Ashukha et al., 2020; Fort et al., 2019).

The described categorization of ensembling methods into three groups, local, semi-local, and global, is not strict. For example, FGE, which is usually assigned to the local group, and SSE, which is usually assigned to the semi-local group, are both based on cyclical learning rate schedules and can be used as either local or semi-local methods, depending on the hyperparameters. Also, local and semi-local methods can be combined with global ones by training several networks from different random initializations, then sampling some networks locally around each of them, and finally ensembling all the obtained networks (Fort et al., 2019). Moreover, recent works on loss landscape analysis hypothesize that two networks trained independently from different random initializations are effectively located in the same basin up to the neuron permutation (Entezari et al., 2022; Ainsworth et al., 2023) and activations renormalization (Jordan et al., 2023), which blurs the boundaries between local and global methods. However, existing local and semi-local methods are not able to find as diverse networks as the global ones can, so in practice, these groups are still easily distinguishable.

If several networks are located in the same basin of low loss, they can be weight-averaged instead of ensembling. The Stochastic Weight Averaging technique (SWA by Izmailov et al. (2018)) averages several checkpoints from one standard training run or models from a local cyclical learning rate schedule method to improve the model quality and robustness.

## 3 Scope of the paper: transfer learning perspective

In the transfer learning setup, a DE for the target task can be trained in two ways: using different separately pre-trained checkpoints to initialize each network or using the same pre-trained checkpoint for all the networks. In terms of the categorization from the previous section, only the former variant can be called global because the networks fine-tuned from the same pre-trained checkpoints are usually located in the same basin of low loss (Neyshabur et al., 2020). Thus, we call the former variant Global DE while the latter is Local DE. Similarly to the relation between local and global methods in regular training, in the transfer learning setup, Global DEs contain more diverse networks and show better quality than Local DEs (Mustafa et al., 2020). At the same time, the difference in training cost between Local and Global DEs is even more pronounced because the pre-training cost is usually huge compared to the fine-tuning cost.

One of the goals of our paper is to understand whether the existing local and semi-local ensembling methods can be beneficial in the transfer learning setup and help close the quality gap between Local and Global DEs. We focus on the situation when only one pre-trained checkpoint is available, as in Local DE, and study the effectiveness of cyclical learning rate methods. This group of methods contains such methods as FGE, SSE, and cSGLD. We choose cyclical methods because they can work in both local and semi-local regimes and show the strongest results in closing the gap with global methods in the regular training setup (Ashukha et al., 2020). In Appendix C, we show that in the transfer learning setup, they are also more effective than other local methods. Cyclical methods have very similar structure and differ mostly in training scheme for the first network and learning rate schedules in the cycles. FGE trains the first network with a more standard long schedule, while SSE and cSGLD use the same schedule in all the cycles. Also, FGE uses short cycles with a triangular learning rate schedule and low maximum learning rate, while SSE and cSGLD use long cycles with a cosine learning rate schedule and relatively high maximum learning rate. In the paper, we mostly experiment with a version of SSE, which fine-tunes the first network with a fixed standard fine-tuning schedule and then trains all the following networks with the same cosine schedule, for which we vary the length and the maximum learning rate. Depending on the cycle hyperparameters, SSE can behave as either a local method (short cycles with low maximum learning rate) or a semi-local method (longer cycles with higher maximum learning rate). We choose to fine-tune the first network in a

fixed way for easier comparison with Local and Global DEs. The results for other variants of cyclical methods are similar and discussed in Appendix D.

To the best of our knowledge, cyclical methods were not previously studied in the transfer learning setup. Moreover, based on our analysis of the SSE behavior, we propose its parallel version, StarSEE, which shows better results in this setup. Several recent works (Mustafa et al., 2020; Wortsman et al., 2022; Rame et al., 2022) study another technique for improving the diversity of Local DEs by training networks using different hyperparameters (learning rate, weight decay, data augmentation, etc.). In Appendix K, we show that this technique can be also effective for Global DEs and StarSSEs, however, it is not able to close the gaps between the methods and change the comparison results.

Since models in a Local DE are usually located in the same basin of low loss, they can be weight-averaged to obtain a single model called a model soup (Wortsman et al., 2022; Rame et al., 2022). If the set of models to average is chosen properly using validation data, a model soup provides a quality boost without the high computational costs of ensembles in the inference stage. Model soups usually show slightly inferior results on the in-distribution data compared to Local DE, but they are more robust to distributional shifts. We also study the weight-averaging potential of SSE and StarSSE in the transfer learning setup, including the robustness of the resulting models to distributional shifts.

## 4 Experimental setup

**Data, architectures, and pre-training.** In most of the experiments, we use a standard ResNet-50 architecture (He et al., 2016) and consider two types of pre-training on the ImageNet dataset (Russakovsky et al., 2015): supervised and self-supervised with the BYOL method (Grill et al., 2020). We use 5 independently pre-trained checkpoints in both cases. As for target tasks, we choose three image classification tasks: CIFAR-10 (Krizhevsky et al., a), CIFAR-100 (Krizhevsky et al., b), and SUN-397 (Xiao et al., 2010). For SUN-397, we use only the first train/test split of the ten specified in the original paper. We also experiment with a CLIP model (Radford et al., 2021) on the ImageNet target task in Section 8, other non-natural image classification target tasks in Appendix N, and Swin-T transformers (Liu et al., 2021) in Appendix M.

**Fine-tuning of individual models.** We replace the last fully-connected layer with a randomly initialized layer having an appropriate number of classes and then fine-tune the whole network using mini-batch SGD with batch size 256, momentum 0.9, and cosine learning rate schedule (Loshchilov and Hutter, 2017). For each target task and each type of pre-training, we choose the optimal number of epochs, weight decay, and initial learning rate for fine-tuning by grid search, allocating a validation subset from the training set (see Appendix B for more detail).

**SSE and StarSSE.** The first network in SSE or StarSSE is always trained in the same manner as an individual fine-tuned network described in the previous paragraph with the optimal number of fine-tuning epochs and maximum learning rate. We denote these optimal hyperparameter values for each task as $\times 1$. For training all the following networks, we use the same learning schedule, for which we vary the number of epochs and maximum learning rate in a cycle. We consider different hyperparameter intervals for different tasks to cover all types of method behavior and denote the values relatively to the optimal ones for individual model fine-tuning (for example, $\times 0.5$ or $\times 2$).

**Aggregating results.** All ensemble accuracy values presented in the plots are averaged over several runs, so mean and standard deviation are shown. For Local and Global DEs, we average over at least 5 runs (we fine-tune 5 networks from each pre-trained checkpoint and average over different pre-trainings and fine-tuning random seeds). For SSE and StarSSE, we average over two runs, each from a different pre-trained checkpoint.

**Connectivity analysis.** To analyze the behavior of ensemble methods, we look at the linear connectivity of ensemble members. Usually, two models are called linearly connected and laying in the same low-loss basin if there is no high-loss barrier on the linear interpolation between them, meaning the loss occurred when linearly connecting the models is lower than the linear interpolation of their losses with the same interpolation coefficients (Entezari et al., 2022). In our analysis, we use accuracy instead of loss. The two metrics behave similarly, however, the absolute accuracy values are easier to interpret. Hence, we examine whether ensemble members lay in the same high-accuracy area (high-accuracy basin) or if there is a zone of lower accuracy on the linear interpolation between them (low-accuracy barrier).

Table 1: Accuracy of ensembles of size 5 of ResNet-50 models with supervised (SV) and self-supervised (SSL) pre-training. Green numbers indicate the gap between Local and Global DE.

| Pre-train. | Dataset | Local DE | Global DE | SSE (opt.) | StarSSE (opt.) |
|---|---|---|---|---|---|
| SV | CIFAR-100 | $85.65_{\pm 0.12}$ | $86.82_{\pm 0.14}$ $(+1.17)$ | $86.04_{\pm 0.18}$ | $86.30_{\pm 0.18}$ |
| | CIFAR-10 | $97.52_{\pm 0.04}$ | $97.69_{\pm 0.05}$ $(+0.17)$ | $97.30_{\pm 0.05}$ | $97.39_{\pm 0.01}$ |
| | SUN-397 | $62.23_{\pm 0.38}$ | $64.65_{\pm 0.10}$ $(+2.42)$ | $63.63_{\pm 0.14}$ | $64.20_{\pm 0.21}$ |
| SSL | CIFAR-100 | $87.33_{\pm 0.21}$ | $88.10_{\pm 0.11}$ $(+0.77)$ | $87.11_{\pm 0.06}$ | $87.63_{\pm 0.12}$ |
| | CIFAR-10 | $97.97_{\pm 0.05}$ | $98.14_{\pm 0.05}$ $(+0.17)$ | $97.94_{\pm 0.04}$ | $98.06_{\pm 0.02}$ |
| | SUN-397 | $65.77_{\pm 0.19}$ | $67.04_{\pm 0.09}$ $(+1.27)$ | $65.79_{\pm 0.10}$ | $66.37_{\pm 0.25}$ |

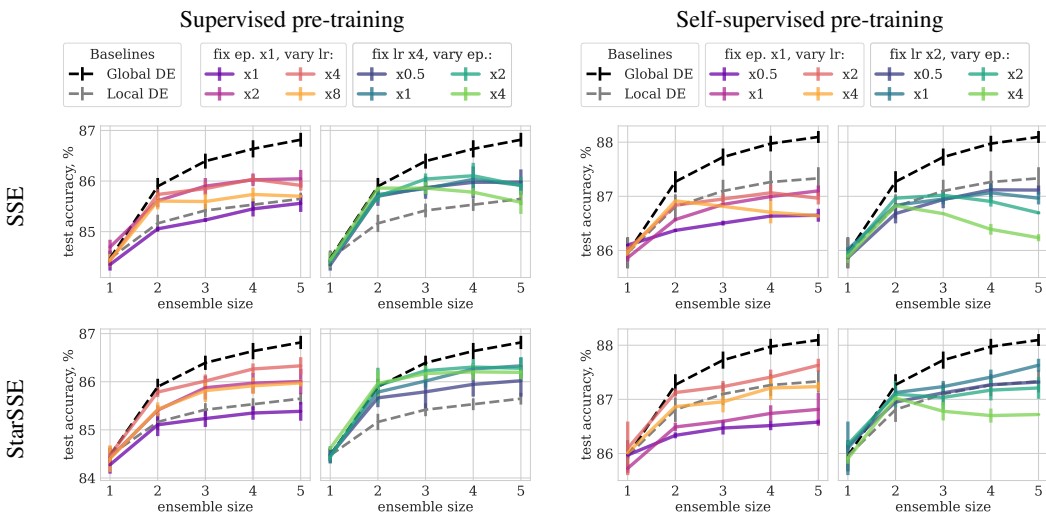

Figure 1: Results of SSEs (top row) and StarSEEs (bottom row) of different sizes on CIFAR-100 for different types of pre-training and varying values of cycle hyperparameters (maximum learning rate and number of epochs). Local and Global DEs are shown for comparison with gray dotted lines.

## 5 Empirical analysis of ensembling in transfer learning

### 5.1 Local and global deep ensembles

As a first step of our empirical analysis of ensembling methods in transfer learning, we compare the quality of Local and Global DEs in our setup. In Table 1, we show the results for ensembles of maximal considered size 5. Also, the results for ensembles of variable sizes are visualized in Figure 1 with gray dashed lines (see Appendix E for the results on other datasets). For both types of pre-training and all considered target tasks, there is a significant quality gap between Local and Global DEs. Thus, the problem of closing this gap is relevant to our setup.

We also confirm that networks from the Local DE lay in the same high-accuracy basin. Gray baseline lines in Figure 2 show accuracy along line segments between random pairs of networks from the same Local DE (see Appendix F for results on other datasets). There are no significant low-accuracy barriers in training and test metrics.

### 5.2 Local and semi-local Snapshot Ensembles

Since a Local DE only explores the pre-train basin, we experiment with the SSE method with varying values of cycle hyperparameters to answer two questions: 1) can we improve a Local DE quality by better exploring the pre-train basin, and 2) we improve it even further by exploring the close vicinity outside the pre-train basin? In this section, we demonstrate the results only on the CIFAR-100 task; the results for other target tasks are similar and presented in Appendix E.

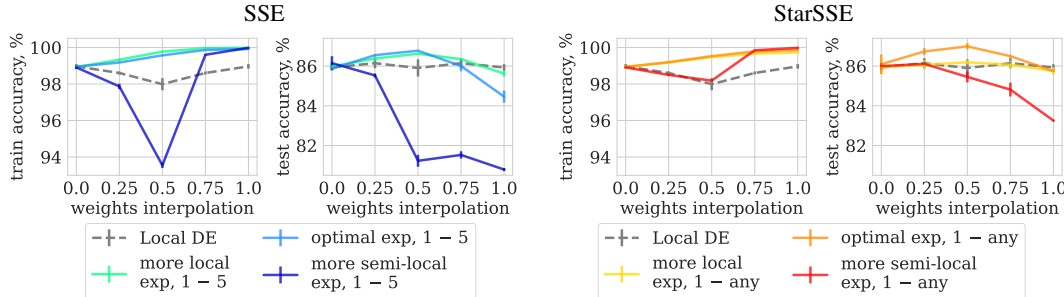

Figure 2: Linear connectivity analysis of Local DEs, SSEs (left plots), and StarSSEs (right plots) on CIFAR-100 with self-supervised pre-training. We show train and test accuracy along line segments between two random networks in Local DEs, between the first and the last (5-th) network in three differently behaving SSEs, and between the first and any other consequent network in three differently behaving StarSSEs. Hyperparameters for more local and more semi-local experiments are the same for SSE and StarSSE, while hyperparameters for the optimal experiments may differ.

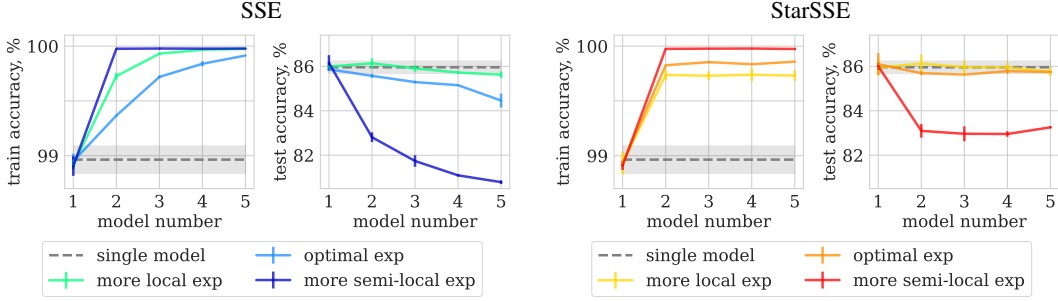

Figure 3: Train and test accuracy of individual models from three differently behaving SSEs (left plots) and StarSSEs (right plots) on CIFAR-100 with self-supervised pre-training (with a single fine-tuned model for comparison). Hyperparameters for more local and more semi-local experiments are the same for SSE and StarSSE, while hyperparameters for the optimal experiments may differ.

We vary the maximum learning rate and number of epochs in SSE cycles in the top row of plots in Figure 1. Varying these hyperparameters similarly changes method behavior because training with a higher learning rate or for a larger number of epochs allows the network to go further from the pre-train checkpoint. For low values of hyperparameters, especially the learning rate, SSE collects very closely located checkpoints; therefore, their ensembling does not substantially improve the prediction quality. When we increase the values, SSE collects more diverse checkpoints, and ensemble quality rises. However, for high hyperparameter values, the ensemble quality significantly degrades with the number of networks due to degradation of the quality of individual models as they move further and further away from the pre-train checkpoint (see next sub-sections).

Overall, SSE with optimally chosen hyperparameters can construct an ensemble of similar, and in some cases even better, quality than Local DE but is not able to close the gap with Global DE (see Table 1). Interestingly, SSE shows strong results for small ensembles, sometimes even matches the Global DE for ensembles of size 2, however, for large ensembles, SSE results become weaker (see Figure 1). SSE shows stronger improvements in the case of supervised pre-training, most likely because of the higher difference in the optimal hyperparameter values for SSE and usual Local DEs.

### 5.3 Analysis of Snapshot Ensemble behavior

To better understand the properties of different types of SSE behavior and the reasons behind them, we analyze three experiments for each setup in more detail. We consider the following three experiments: the *more local* experiment (with low learning rate and very slow-growing ensemble quality), the *more semi-local* experiment (with high learning rate, a large number of epochs, and ensemble quality degradation for large ensemble sizes), and the *optimal* experiment (with medium learning rate and

number of epochs, and the best quality for an ensemble of size 5). See Appendix B for the exact values of hyperparameters for these three experiments. In this section, we demonstrate the results only for experiments with self-supervised pre-training and the CIFAR-100 dataset. The results with supervised pre-training are similar and presented in Appendix F.

In the left pair of plots in Figure 2, we show how train and test accuracy behave along the line segment between the first and the last (5-th) network in the SSE. Metric values along the line segment between two random networks from the same Local DE are shown as baselines. From the loss landscape perspective, more local and more semi-local experiments indeed demonstrate the expected local and semi-local behavior, respectively. Interestingly, the behavior of optimal experiments is also local, without low-accuracy barriers. These results confirm that SSE can behave as both local and semi-local methods and that its local variant provides the best results. In Appendix G, we provide additional locality analysis based on the accuracy behavior on 2D planes in the weight space.

In the left pair of plots in Figure 3, we show how train and test accuracy of individual models in SSE changes with each cycle. In all the experiments, model test accuracy degrades with each cycle, even though the train accuracy increases or stays at the maximum value. From this, we can conclude that the further individual models walk away from the pre-train checkpoint, the more they overfit the training data of the target task and, as a result, lose the advantages of transfer learning.

SSE with optimal hyperparameters can improve over Local DE by better exploring the pre-train basin and collecting more diverse networks (see Appendix J for the detailed diversity analysis). The diversity of these networks even makes up for the gradual decrease in individual model quality. However, leaving the pre-train basin with SSE results in the most pronounced degradation of individual model test accuracy, which leads to decreasing ensemble quality. We also hypothesize that due to the decrease in the quality of individual models, SSE with any values of parameters is not able to effectively construct large ensembles in the transfer learning setup (any SSE plot will start to decrease if we grow the ensemble size further).

Whether staying in the pre-train basin is an essential condition to take advantage of knowledge transfer and obtain high-quality ensembles or the considered methods explore the vicinity of the pre-train basin in a non-effective way is an interesting question for future research. We suspect that the main reason why SSE can not leave the pre-train basin without losing the transfer learning advantage is the fact that SSE only looks at the target task loss landscape without taking into consideration the source task one. Shwartz-Ziv et al. (2022) show that accounting for the source loss landscape shape around the pre-trained checkpoint can improve fine-tuning accuracy of individual models and local ensembles. However, to do so, they fit a normal prior on the source task, centered in the pre-trained checkpoint, which most likely can be effective only locally in the pre-train basin (see Appendix L for additional experiments on SSE with similar regularizations). Hence, more advanced techniques should be developed to account for the source loss landscape shape outside of the pre-train basin.

## 5.4 StarSSE — a better version of Snapshot Ensemble for transfer learning

Our study of SSE behavior shows that, in the transfer learning setup, standard cyclical learning rate methods suffer from individual network quality degradation due to moving too far from the pre-train checkpoint and overfitting to the target training data, and this effect becomes more pronounced with each cycle. SSEs with long cycles and high learning rates are affected the most by this, but at the same time, they show high quality for small ensemble sizes (see results for ensembles of size 2 at the top row of Figure 1). So, even though the second network found by SSE is weaker as an individual model, it may be a better candidate for the ensemble with the first one than another independently fine-tuned network, as in Local DE.

Based on these observations, we propose a parallel version of SSE, StarSSE, which is more suitable for the transfer learning setup. StarSSE trains the first model with the standard fine-tuning schedule, as in Local DE or SSE. Then, all the following models are trained with the same schedules as in SSE cycles but in a parallel rather than sequential manner. Each of these models is initialized with the weights of the first fine-tuned model. As a result, we have a "star" with the first model in the center and all other models at the ends of the "rays". Due to the parallel structure, StarSSE does not move further from the pre-trained checkpoint with each training cycle as SSE, but at the same time, it explores the area with good candidates for ensembling around the first fine-tuned network.

We show how StarSSE with different cycle hyperparameters behaves in the bottom row of plots in Figure 1 (for consistency with SSE, we call hyperparameters for training all the networks except the first one as cycle hyperparameters). The results on other datasets are presented in Appendix E. StarSSE demonstrates more stable results for medium and high values of cycle hyperparameters than SSE, and as a result, it produces better larger size ensembles. Even though for the highest considered hyperparameter values, StarSSE quality still decreases with increasing ensemble size, its degradation is much less severe. As can be seen from Table 1, the optimal StarSSE outperforms the optimal SSE.

To better understand the difference between StarSSE and SSE behavior, we conduct the same analysis of linear connectivity and individual model quality for three distinct StarSSE experiments, as in the previous section for SSE. For easier comparison, we choose the same cycle hyperparameters for more local and more semi-local experiments as for their SSE counterparts. For the optimal experiment, we choose its own hyperparameter values to compare the best results of the methods. The exact values of hyperparameters can be found in Appendix B. In the right pair of plots in Figure 2, we show how train and test accuracy of StarSSE behave along the line segment between the first and any of the following models (it does not matter which model to choose because all of them are trained independently in the same manner). The more semi-local StarSSE experiment demonstrates much more local behavior from the loss landscape perspective than the SSE one, which confirms that StarSSE does not go too far from the pre-train basin, even with relatively high hyperparameter values. The optimal StarSSE experiment demonstrates local behavior; therefore, StarSSE, similarly to SSE, should be used only as a local method in the transfer learning setup. As shown in the right pair of plots in Figure 3, all the models in StarSSE, starting from the second one, have a very similar quality, which makes StarSSE more stable for larger ensemble sizes. For the optimal experiment, the quality degradation between the first and the following models is very low and can be compensated by the model diversity in StarSSE to achieve higher results than Local DE (see Appendix J for detailed diversity analysis).

In conclusion, StarSSE is a more effective alternative to cyclical learning rate methods for the transfer learning setup, which can work with higher values of cycle hyperparameters to obtain diverse models without gradual degradation of individual model quality. StarSSE is technically very similar to Local DE, however, there is an important conceptual difference between them. StarSSE separates the two processes: 1) model fine-tuning from the pre-train checkpoint to a high-quality area for the target task, and 2) exploration of this area for obtaining diverse models. On the contrary, Local DE tries to do both of them jointly each time it fine-tunes a network. The design of StarSSE allows it to make exploration with higher hyperparameter values, which are suboptimal for the initial fine-tuning process, and obtain a more diverse set of models. The fact that optimal hyperparameter values for StarSSE are higher than the optimal $\times 1$ values for Local DE (see Figure 1) supports this reasoning.

## 6    Soups of SSE and StarSSE models

In the previous section, we show that SSE and StarSSE methods can be effective in transfer learning but only in constructing local ensembles. Even though the locality usually leads to limited model diversity, it also has its benefits: it allows to weight-average a set of models to obtain a single, more accurate, and robust model. To constitute a strong weight-averaged model, a set of models needs to lie in the same low-loss basin and have a convex loss landscape between the models. Diversifying the models in a set can improve the averaging results; therefore, model soups (Wortsman et al., 2022; Rame et al., 2022), which average independently fine-tuned models, show better results than SWA (Izmailov et al., 2018), which averages several checkpoints from the same training run. Moreover, for even higher diversity, model soups are usually constructed from the models fine-tuned with variable hyperparameters. However, the relation between the final quality and the diversity of the models is not as straightforward for weight-averaging as for ensembles because diversifying models too much may result in a non-convex loss between them. Because of this, averaging all fine-tuned models into a uniform model soup does not result in a high-quality model, and, to constitute an effective greedy model soup, a subset of models needs to be chosen using the validation data.

Linear connectivity analysis in Figure 2 shows that averaging two models from the Local DE does not lead to a strong model, while models from optimal SSE and StarSSE experiments lay in a more convex part of the pre-train basin and average much better. This observation motivates us to investigate the weight-averaging of models from SSE or StarSSE to construct uniform model soups. We compare ensembles and uniform model soups of models obtained with Local DE, SSE, and StarSSE in the top row in Figure 4 (see Appendix H for results with supervised pre-training). Firstly, weight-averaging

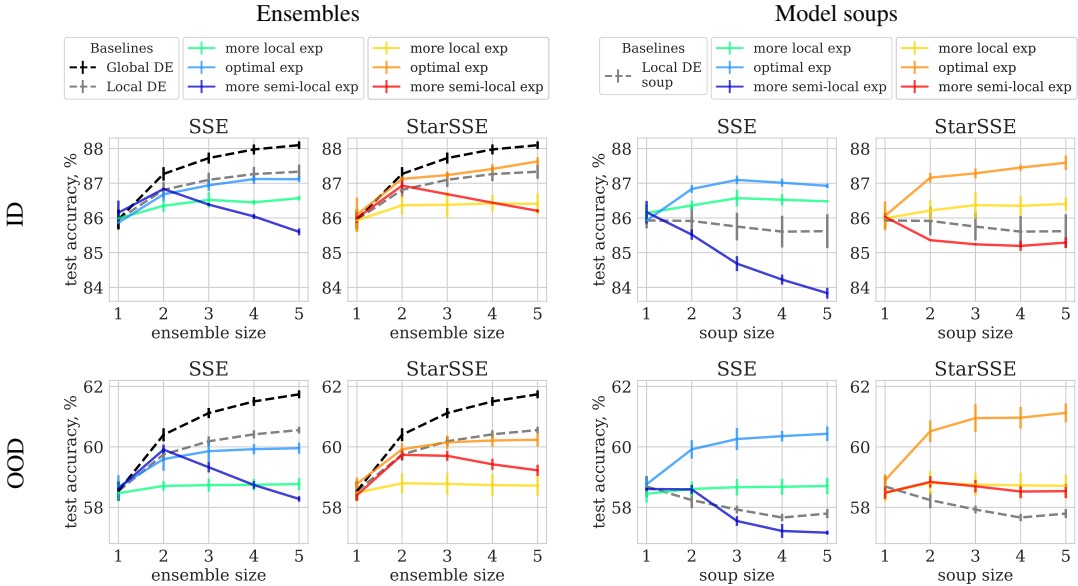

Figure 4: Results of ensembles (left plots) and model soups (right plots) of different sizes on ID (CIFAR-100 test set, top row) and OOD (CIFAR-100C, bottom row) for SSE and StarSSE with self-supervised pre-training. For OOD, we measure the average accuracy over all possible corruptions and severity values. Standard deviations are calculated over different pre-training checkpoints and/or fine-tuning random seeds.

Local DE models does not improve the quality over individual models. Model soups from more local and optimal SSE and StarSSE experiments show better results than an individual fine-tuned model. However, for the more local experiments, the improvement is minor due to the low model diversity. Weight-averaging networks from the more local SSE in our setup is very close to standard SWA. Models from more semi-local SSE and StarSSE experiments can not constitute adequate model soups due to the low-accuracy barriers between the models. Comparison between SSE and StarSSE optimal experiments shows the advantage of the latter: the soup of StarSSE models demonstrates better results than from SSE ones and matches the quality of the ensemble of the same models.

To conclude, StarSSE can collect a set of models in the pre-train basin, which are diverse enough to constitute a strong local ensemble and, at the same time, are located in a convex part of the basin and result in a strong uniform model soup after weight-averaging. In standard model soups, models are usually fine-tuned with varying hyperparameters for higher diversity, and StarSSE can also benefit from this hyperparameter diversification (see Appendix K). This makes the further analysis of their combination a promising direction for future research.

## 7 Robustness analysis

One of the main benefits of ensembles and model soups is their robustness to different types of test data modifications. In this section, we evaluate the robustness properties of the ensemble and soup variants of SSE and StarSSE on the corrupted version of the CIFAR-100 dataset. CIFAR-100C dataset (Hendrycks and Dietterich, 2019) includes 19 synthetic corruptions (e. g., Gaussian noise or JPEG compression), each having 5 different severity values. We compare the performance of all the methods on the in-distribution CIFAR-100 (ID) and the out-of-distribution CIFAR-100C (OOD) in Figure 4 (see Appendix I for results with supervised pre-training). For SSE and StarSSE, we consider the same three differently behaving experiments that are used in the previous sections.

First of all, we observe that more local ensembles lose their potential on OOD completely and have near-constant accuracy for all the considered ensemble sizes. In contrast, the behavior of more semi-local ensembles is very similar on ID and OOD in comparison to the Local DE. This difference indicates that model diversity is even more important under domain shifts. The quality of optimal

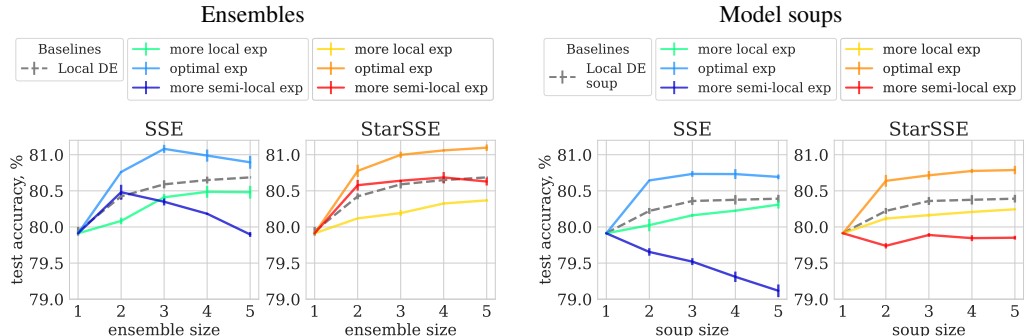

Figure 5: Results of ensembles (left plots) and model soups (right plots) of CLIP models on ImageNet. Three differently behaving SSE and StarSSE experiments are shown with Local and Global DEs for comparison. Hyperparameters for more local and more semi-local experiments are the same for SSE and StarSSE, while hyperparameters for the optimal experiments may differ.

ensembles also decreases compared to a Local DE, agreeing with their local behavior. Overall, ensemble variants of SSE and StarSSE are less robust than DEs to the test data corruption. We discuss the reasons for such behavior in more detail in Appendix J.

For model soups, we observe similar changes in the behavior of more local and more semi-local experiments. The former can not provide a better quality model on OOD, while the results of the latter even improve in comparison to the soup of Local DE models. Thus, model diversity matters for the soups too. Interestingly, optimal soups significantly outperform optimal ensembles, while on the clean data, these two paradigms show almost the same results. This observation matches with the findings of Wortsman et al. (2022), who claim the superiority of standard model soups over ensembles under distribution shifts. Overall, the best results on OOD data are achieved with an optimal soup of StarSSE models (of those trained from a single pre-trained checkpoint). It not only outperforms its ensemble variant but also provides considerably better results than Local DE. Thus, a soup of StarSSE models is a good option for fine-tuning a robust model given a single pre-trained checkpoint.

## 8    Large-scale experiments

Our main analysis of SSE and StarSSE methods is focused on moderate-size networks and smaller target tasks to be able to accurately compare the results with both Local and Global DE baselines. However, to confirm that the main conclusions of the paper stand in more practical large-scale setups, we experiment with the CLIP ViT-B/32 model pre-trained on the large dataset of the image-text pairs and the ImageNet target task. In these experiments, we drop the Global DE baseline due to its extremely high computational cost. In Figure 5, we compare the results of three differently behaving SSEs and StarSSEs (see Appendix B for specific hyperparameter values) and the Local DE baseline. SSE and StarSSE with optimal hyperparameter values significantly outperform Local DE both as ensembles and as uniform model soups. The comparison of StarSSE and SSE confirms previously discussed benefits of the parallel training strategy: StarSSE ensembles and model soups, trained with high hyperparameter values, show less quality degradation than their corresponding SSE counterparts. We conclude that StarSSE is effective for constructing better local ensembles in practical large-scale setups too.

## 9    Conclusion

In this work, we study the effectiveness of cyclical learning rate methods for training ensembles from a single pre-trained checkpoint in the transfer learning setup. We demonstrate that while better exploration of the pre-train basin with these methods can be beneficial, they can not leave the basin without losing the advantages of transfer learning and significant quality degradation. Based on our analysis, we also propose a parallel version of the SSE method for transfer learning, StarSSE. Models from StarSSE not only constitute a better ensemble but also lay in a more convex part of the pre-train basin and can be effectively combined into a high-quality model soup.

## Acknowledgments and Disclosure of Funding

We would like to thank Maxim Kodryan, Arsenii Kuznetsov, and the anonymous reviewers for their valuable comments. Also, we are very grateful to Polina Kirichenko and Timur Garipov for presenting the paper at the NeurIPS poster session, which we could not attend, unfortunately. The results on ResNet with self-supervised pre-training were obtained by Ildus Sadrtdinov with the support of the grant for research centers in the field of AI provided by the Analytical Center for the Government of the Russian Federation (ACRF) in accordance with the agreement on the provision of subsidies (identifier of the agreement 000000D730321P5Q0002) and the agreement with HSE University №70-2021-00139. The empirical results were supported in part through the computational resources of HPC facilities at HSE University (Kostenetskiy et al., 2021).

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

# A   Limitations and societal impact

The main limitation of our work is choosing the hyperparameter values for Local and Global DEs based on the validation accuracy of individual models. Even though it is standard practice for DEs, such hyperparameters may lead to suboptimal ensemble quality. However, we used the same hyperparameters for the training of the first SSE and StarSSE models, so their quality may also be suboptimal. Also, an extension of the study to other domains may be an interesting addition to our work.

To the best of our knowledge, our work does not have any negative societal impact, except the substantial computational costs of the conducted study. However, our work improves understanding of the ensembles and model soups construction using only one pre-trained checkpoint, which can help to reduce computational costs in future projects.

# B   Experimental setup details

All the described experiments were carried out using the PyTorch framework (Paszke et al., 2019).

**Pre-training of ResNets.** For the supervised case, we use available ResNet-50 checkpoints[2] pre-trained by Ashukha et al. (2020). We take five of them trained from different random initializations with the same hyperparameters. For the self-supervised case, we also pre-train five checkpoints from scratch for 1000 epochs using the hyperparameters from the original BYOL paper. We use the *solo-learn* library for BYOL pre-training (da Costa et al., 2022).

**Swin-T.** We pre-train five Swin-T checkpoints from scratch in a supervised manner on ImageNet for 300 epochs. We use the official code and hyperparameters from the original Swin paper. For fine-tuning of individual models for Local and Global DEs, we use AdamW (Loshchilov and Hutter, 2019) and linear learning rate warm-up for $10\%$ of the fine-tuning epochs. For SSE and StarSSE, we also use AdamW for each cycle but the learning rate warm-up is applied only in the first cycle.

**Fine-tuning hyperparameters grid search for ResNet and Swin-T models.** We divide the training data into training and validation subsets in 1 to 10 ratio for CIFAR10, CIFAR100 and in 1 to 5 ratio for SUN397 as in the work of Kornblith et al. (2019) and 1 to 5 ration for Clipart and Chest-X. We use grid search for the values of training epochs, initial learning rate and weight decay giving the best quality of individual fine-tuned networks on validation subsets. For ResNet-50 (both types of pre-training), we consider weight decays from $[2 \cdot 10^{-5}, 1 \cdot 10^{-4}, 5 \cdot 10^{-4}]$ and initial learning rates from $[5 \cdot 10^{-4}, 10^{-3}, 2.5 \cdot 10^{-3}, 5 \cdot 10^{-3}, 10^{-2}, 2.5 \cdot 10^{-2}, 5 \cdot 10^{-2}, 10^{-1}, 2 \cdot 10^{-1}]$. For Swin-T, the ranges are $[10^{-2}, 5 \cdot 10^{-2}, 2.5 \cdot 10^{-1}]$ and $[2.5 \cdot 10^{-5}, 5 \cdot 10^{-5}, 10^{-4}, 2.5 \cdot 10^{-4}, 5 \cdot 10^{-4}, 10^{-3}]$, respectively. As for the number of training epochs, we determine the value corresponding to 20000 SGD steps (this procedure is taken from Kornblith et al. (2019); the value differs across datasets, making up 102 for CIFAR-10, CIFAR-100, 256 for SUN-397, 8 for Chest-X and 19 for Clipart) and consider its multipliers from the list $[\times 0.03125, \times 0.0625, \times 0.125, \times 0.25, \times 0.5, \times 1]$ rounded down. The resulting optimal sets of hyperparameters are shown in Table 2.

**CLIP.** We use a CLIP ViT-B/32 model pre-trained by the authors on a large dataset of image-text pairs. For fine-tuning of individual models for Local DEs, we follow Wortsman et al. (2022) using AdamW (Loshchilov and Hutter, 2019), linear learning rate warm-up for 500 SGD steps and the best combination of the training hyperparameters from their random search (number of epochs 11, batch size 512, learning rate $1.23 \cdot 10^{-5}$, label smoothing 0.134, and augmentation Rand-Augment with $M = 19, N = 0$). For SSE and StarSSE, we also use AdamW, learning rate warm-up, and the same batch size, label smoothing, and augmentations in each cycle.

**Hyperparameters of more local/optimal/more semi-local experiments.** The hyperparameter multipliers for more local, SSE optimal, StarSSE optimal and more semi local experiments are presented in Table 3.

**Amount of computations.** We use NVIDIA TESLA V100 and A100 GPUs for computations in our experiments. Total expenses on pre-training constitute 8K GPU hours for self-supervised ResNet-50 (BYOL) and supervised Swin-T. Fine-tuning models for Local and Global DEs and cyclical learning

---

[2]https://github.com/SamsungLabs/pytorch-ensembles

Table 2: Optimal values of training epochs, initial learning rate and weight decay for fine-tuning an individual model.

| Pre-train. | Target dataset | epochs | lr | wd |
|---|---|---|---|---|
| **SV ResNet-50** | CIFAR-10 | 51 | $2.5 \cdot 10^{-2}$ | $2 \cdot 10^{-5}$ |
| | CIFAR-100 | 25 | $5 \cdot 10^{-3}$ | $5 \cdot 10^{-4}$ |
| | SUN-397 | 256 | $5 \cdot 10^{-4}$ | $5 \cdot 10^{-4}$ |
| | ChestX | 8 | $2.5 \cdot 10^{-2}$ | $5 \cdot 10^{-4}$ |
| | Clipart | 19 | $2.5 \cdot 10^{-2}$ | $5 \cdot 10^{-4}$ |
| **SSL ResNet-50** | CIFAR-10 | 25 | $5 \cdot 10^{-2}$ | $10^{-4}$ |
| | CIFAR-100 | 25 | $5 \cdot 10^{-2}$ | $10^{-4}$ |
| | SUN-397 | 16 | $10^{-1}$ | $10^{-4}$ |
| | ChestX | 8 | $2 \cdot 10^{-1}$ | $5 \cdot 10^{-4}$ |
| | Clipart | 19 | $10^{-1}$ | $2 \cdot 10^{-5}$ |
| **SV Swin-T** | CIFAR-100 | 102 | $10^{-4}$ | $2.5 \cdot 10^{-1}$ |
| | SUN-397 | 256 | $2.5 \cdot 10^{-5}$ | $5 \cdot 10^{-2}$ |

Table 3: Considered multipliers of the learning rate and the number of epochs for more local, SSE optimal, StarSSE optimal, and more semi-local experiments for different datasets and types of pre-training.

| Pre-train. | Target dataset | *more local* | | *SSE optimal* | | *StarSSE optimal* | | *more semi-local* | |
|---|---|---|---|---|---|---|---|---|---|
| | | lr | ep. | lr | ep. | lr | ep. | lr | ep. |
| **SV ResNet-50** | CIFAR-10 | ×0.25 | ×1 | ×2 | ×0.25 | ×2 | ×0.5 | ×2 | ×2 |
| | CIFAR-100 | ×1 | ×1 | ×2 | ×1 | ×4 | ×1 | ×8 | ×4 |
| | SUN-397 | ×0.5 | ×1 | ×32 | ×0.125 | ×32 | ×0.125 | ×32 | ×1 |
| | ChestX | ×0.25 | ×1 | ×4 | ×2 | ×2 | ×2 | ×8 | ×4 |
| | Clipart | ×0.25 | ×0.5 | ×1 | ×0.5 | ×2 | ×0.5 | ×4 | ×2 |
| **SSL ResNet-50** | CIFAR-10 | ×0.5 | ×1 | ×1 | ×1 | ×2 | ×1 | ×4 | ×4 |
| | CIFAR-100 | ×0.25 | ×1 | ×2 | ×0.5 | ×4 | ×0.5 | ×4 | ×4 |
| | SUN-397 | ×0.5 | ×1 | ×1 | ×1 | ×2 | ×1 | ×4 | ×8 |
| | ChestX | ×0.5 | ×1 | ×1 | ×1 | ×1 | ×2 | ×2 | ×4 |
| | Clipart | ×0.5 | ×0.5 | ×4 | ×0.5 | ×4 | ×0.5 | ×8 | ×2 |
| **SV Swin-T** | CIFAR-100 | ×0.5 | ×0.25 | ×2 | ×0.25 | ×4 | ×0.5 | ×4 | ×1 |
| | SUN-397 | ×1 | ×1 | ×16 | ×0.125 | ×16 | ×0.25 | ×16 | ×1 |
| **CLIP ViT-B/32** | ImageNet | ×1 | ×0.25 | ×2 | ×1 | ×2 | ×1 | ×6 | ×1 |

rate ensembles took another 17K GPU hours, resulting in a total amount of approximately 25K GPU hours.

## C   Non-cyclical local ensemble methods

In this section, we experiment with non-cyclical local ensemble methods in the transfer learning setup and show that they are generally less effective than the cyclical ones. We choose three non-cyclical methods: KFAC-Laplace (Ritter et al., 2018), SWAG (Maddox et al., 2019) and SPRO (Benton et al., 2021). KFAC-Laplace is the most simple one and does not require any additional training. Given one usually trained network, it computes the Laplace approximation with a block Kronecker

Table 4: Accuracy and diversity of ensembles of size 5 for non-cyclical local methods, Local DE, and optimal SSE and StarSSE. Self-supervised pre-training.

| Dataset | | Local DE | SSE (optim.) | StarSSE (optim.) | KFAC Laplace | SPRO | SWAG |
|---|---|---|---|---|---|---|---|
| CIFAR-100 | *acc.* | $87.33_{\pm0.21}$ | $87.11_{\pm0.06}$ | $87.63_{\pm0.12}$ | $86.01_{\pm0.02}$ | $86.46_{\pm0.08}$ | $86.99_{\pm0.08}$ |
| | *div.* | $66.25_{\pm3.56}$ | $71.57_{\pm5.94}$ | $71.51_{\pm2.08}$ | $35.38_{\pm2.07}$ | $33.94_{\pm16.96}$ | $36.74_{\pm5.86}$ |
| SUN-397 | *acc.* | $65.77_{\pm0.19}$ | $65.79_{\pm0.10}$ | $66.37_{\pm0.25}$ | $64.39_{\pm0.07}$ | $64.71_{\pm0.09}$ | $65.83_{\pm0.11}$ |
| | *div.* | $47.41_{\pm2.48}$ | $57.69_{\pm6.51}$ | $60.74_{\pm1.94}$ | $17.87_{\pm0.94}$ | $24.37_{\pm9.87}$ | $49.45_{\pm5.70}$ |

factored (KFAC) estimate of the Hessian matrix and then samples network weights from the Gaussian distribution with the trained network as a mean and the obtained KFAC estimate as a covariance matrix. SWAG also constructs the Gaussian distribution but uses the SWA solution as a mean and approximates the covariance matrix based on the SWA checkpoints. SWAG is shown to be the most effective alternative to cyclical methods in the regular training setup (Ashukha et al., 2020). SPRO shows competitive results to SWAG by constructing a simplex in the weight space with the given trained network as one of the vertexes and then sampling ensemble members from it.

In the experiments, we consider ResNet-50 with self-supervised pre-training on CIFAR-100 and SUN-397 datasets. We use the official SWAG repository[3] for the implementations of KFAC-Laplace and SWAG, and our own implementation of SPRO. We use grid search to determine the optimal hyperparameters for the non-cyclical local methods:

- **KFAC-Laplace.** We search for the optimal sampling scale from the list $[0.2, 0.4, 0.6, 0.8, 1.0, 1.2]$. The resulting values are $1.0$ for CIFAR-100 and $0.6$ for SUN-397.

- **SWAG.** We continue training the fine-tuned checkpoints with a constant learning rate ($0.08$ for CIFAR-100 and $0.16$ for SUN-397) for 19 more epochs, gathering new models at the end of each epoch (overall, there are 20 models counting the fine-tuned one). Then, we search for the value of sampling scale from the list $[0.1, 0.2, 0.4, 0.6, 0.8, 1.0, 1.2]$ similar to KFAC-Laplace. The optimal values are $0.1$ for CIFAR-100 and $0.2$ for SUN-397.

- **SPRO.** We start with a fine-tuned checkpoint and train 2 more points to form 2-simplexes. We vary the learning rate ($[0.0125, 0.025, 0.05, 0.1]$) and the number of epochs (CIFAR-100: $[5, 10, 20]$, SUN-397: $[4, 8, 16]$), as well as the regularization coefficient ($[10^{-6}, 10^{-5}]$) to train these points. The optimal tuples are $(0.05, 10, 10^{-5})$ for CIFAR-100 and $(0.05, 16, 10^{-6})$ for SUN-397.

In Table 4, we compare accuracy and diversity of 5-network ensembles obtained with the considered non-cyclical local methods, Local DE baseline, and optimal SSE and StarSSE (see Appendix J for the definition of the diversity metric). KFAC-Laplace and SPRO show inferior accuracy and diversity compared to both Local DE and SSE; hence, these methods' behavior is extremely local. SWAG demonstrates stronger results, even comparable to Local DE and SSE on SUN-397. However, StarSSE is more effective than SWAG on both datasets. In conclusion, even though cyclical methods can not benefit from the exploration of several low-loss basins in transfer learning setup, they are still preferable to other local methods, similarly to the regular training setup. Based on this analysis, we decided to focus on cyclical methods in the paper.

## D   Variants of cyclical ensemble methods

In this section, we discuss other variants of cyclical learning rate methods and explain our choice of SSE with a fixed first model fine-tuning schedule for the main study. In all the experiments, we consider ResNet-50 on CIFAR-100 with self-supervised pre-training.

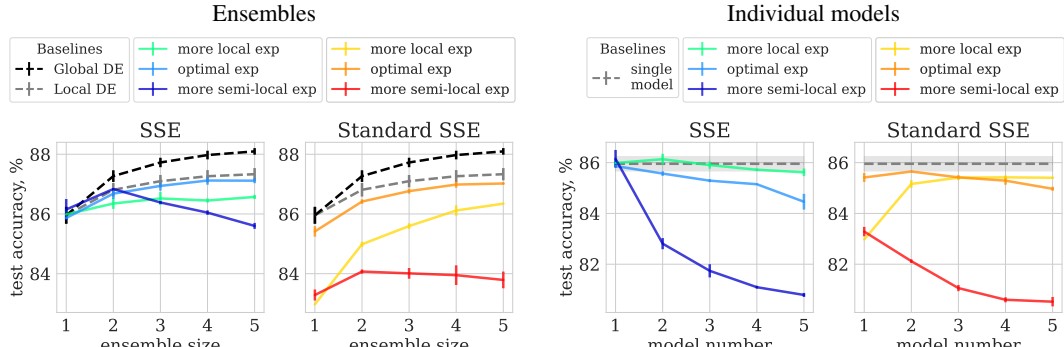

Figure 6: Results of ensembles (left plots) and individual models (right plots) for our variant of SSE with a fixed schedule for the first network fine-tuning and Standard SSE, which uses the same hyperparameters in each cycle. CIFAR-100, self-supervised pre-training. Hyperparameters for all three differently behaving experiments are the same for both methods.

## D.1 SSE vs Standard SSE: why we fine-tune the first model with a fixed schedule

Standard SSE (Huang et al., 2017) trains all the models using the same learning rate schedule in each cycle. However, we choose to use a separate fixed schedule for the first model fine-tuning. This separation allows us to experiment with different hyperparameters for loss landscape exploration without harming the first fine-tuning cycle, which is responsible for moving from the pre-trained checkpoint to the high-quality area in the weight space for the target task.

In the right pair of plots in Figure 6, we compare the behavior of individual model quality from our variant of SSE and the standard SSE with the same hyperparameter values (we use the values for three differently behaving SSEs specified in Appendix B). The first model from the optimal standard SSE experiment has adequate quality, which can also be improved a bit by choosing hyperparameter values optimal for Local DE instead of the ones optimal for our SSE. Since hyperparameter values of the more local and the more semi-local experiments are very different from the hyperparameters usually used for individual model fine-tuning in Local DEs, the quality of the first model from standard SSE is very low in these cases. Moreover, going through more cycles does not help due to slow quality improvement for low hyperparameter values and progressive quality degradation for high hyperparameter values. As a result, the ensemble quality of standard SSE with different hyperparameter values varies a lot, not only due to changes in exploration but also due to different fine-tuning quality in the first place (see the left pair of plots in Figure 6).

To solve the described problem, the first low-quality models are usually dropped from the standard SSE. However, this can not help in the case of high hyperparameter values, which is important to us in this paper. Moreover, for different hyperparameter values, we would need to drop a different number of first models, which would make the comparison of different SSEs and DEs much less clear.

## D.2 cSGLD

cSGLD (Zhang et al., 2020) technically is very similar to SSE and uses the same cyclical cosine learning rate schedule. However, instead of taking only one final model from each cycle, cSGLD samples several models on the last epochs using Langevin dynamics. In our experiments, we compare SSE and cSGLD, which use separate hyperparameters for the first cycle due to the conclusion from the previous section. All other aspects of the cSGLD procedure are the same as in the original paper, which includes the number of sampled models in each cycle equal to 3.

In the right pair of plots in Figure 7, we show that models from the same cycle of cSGLD are located in the same basins. Hence, exploration of the weight space mostly depends on obtaining the models from different cycles, which can indeed end up in different basins, similarly to the SSE ones (see the left pair of plots in Figure 7). Based on these observations, we compare SSE and cSGLD in

---

[3]`https://github.com/wjmaddox/swa_gaussian/tree/master`

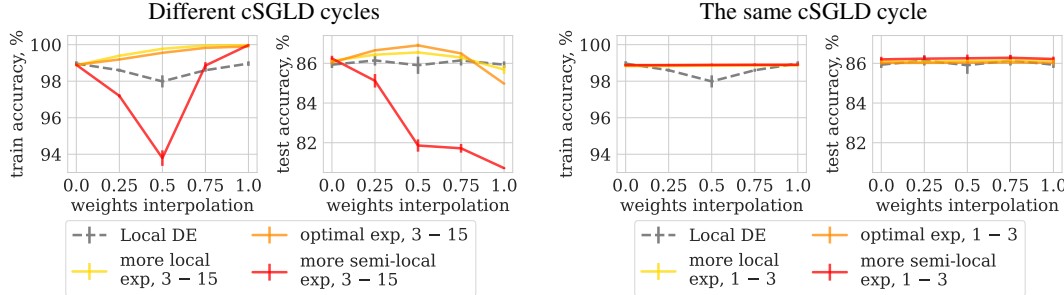

Figure 7: Linear connectivity analysis of cSGLD on CIFAR-100 with self-supervised pre-training. We show train and test accuracy along line segments between two random networks in Local DEs, between the last network of the first cycle (#3) and the last network of the last cycle (#15) of cSGLD (left plots), and between the first (#1) and the last network (#3) from the first cycle of cSGLD (right plots).

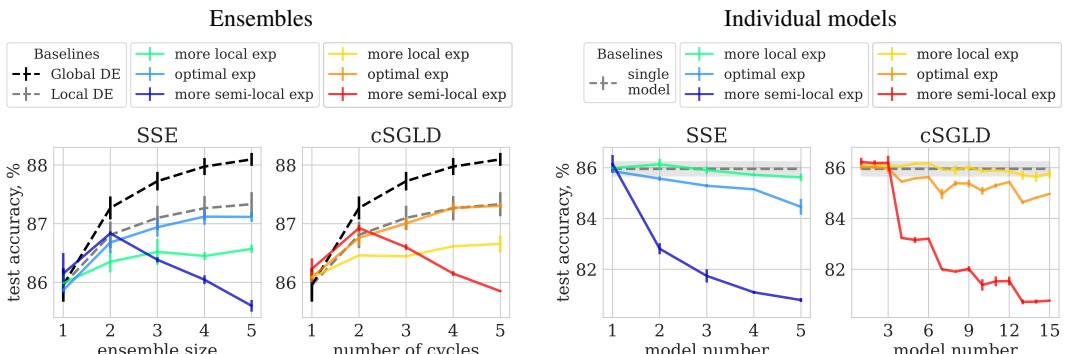

Figure 8: Results of ensembles (left plots) and individual models (right plots) for SSE and cSGLD. CIFAR-100, self-supervised pre-training. Hyperparameters for all three differently behaving experiments are the same for both methods. Every cycle of cSGLD adds 3 new models, so cSGLD ensembles have 3 times more models than SSE, resulting in a slightly better quality.

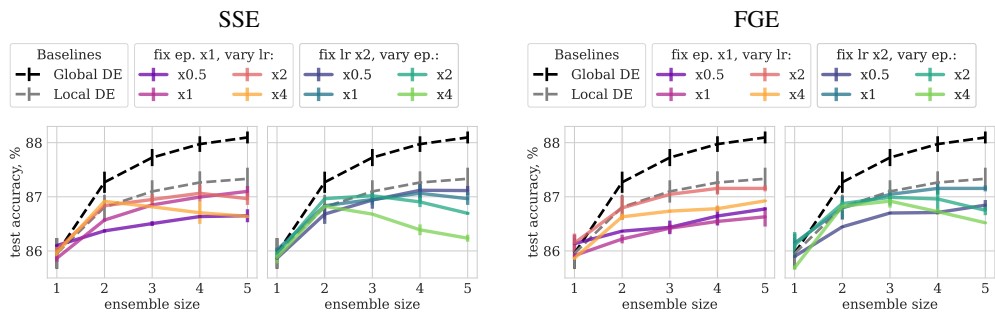

Figure 9: Results of SSEs (left plots) and FGEs (right plots) of different sizes on CIFAR-100 with self-supervised pre-training for varying values of cycle hyperparameters (maximum learning rate and a number of epochs). Local and Global DEs are shown for comparison with gray dotted lines.

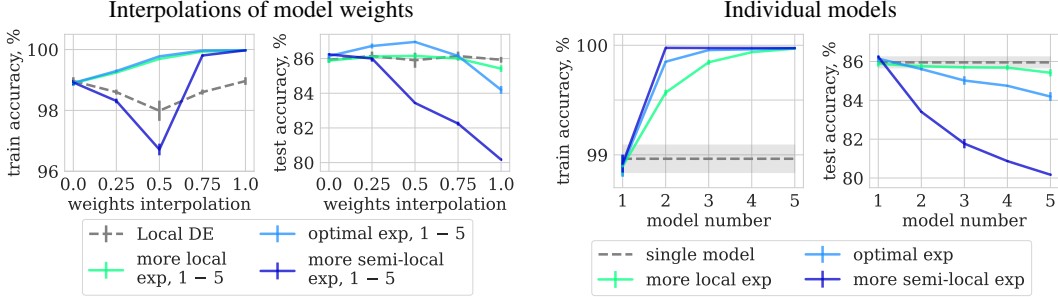

Figure 10: Linear connectivity analysis (left plots) and individual model quality (right plots) of three differently behaving FGEs on CIFAR-100 with self-supervised pre-training. On the left plots, we show train and test accuracy along line segments between two random networks in Local DEs and between the first and the last (5-th) networks in FGEs.

Figure 8, fixing the number of cycles instead of the number of models in an ensemble. We conclude that cSGLD behaves very similarly to SSE and choose SSE for the main study because it gives almost the same quality using fewer networks.

### D.3 FGE

FGE (Garipov et al., 2018) uses a triangular schedule instead of the cosine one for each cycle and trains the first model with a separate schedule in the same manner as we do in our version of SSE. In our experiments, we train the first model of FGE with the same hyperparameters as the first model of SSE.

In Figure 9, we show the results of FGE with different cycle hyperparameters and compare them to the SSE ones. FGE experiments show similar types of behaviors, as we discussed for SSE in the main text. However, due to the triangular learning rate schedule, FGE behaves more locally than SSE with the same hyperparameter values. We also analyze the loss along line segments between the models from FGE and their individual quality in Figure 10. The results show that FGE can work in local and semi-local regimes, and it achieves the optimal ensemble quality in the local one.

## E    Ensembling results of SSE and StarSSE

In this section, we provide additional results of SSE and StarSSE on CIFAR-10 and SUN-397 datasets. Figure 11 is complementary to Figure 1 from the main text and compares SSE and StarSSE with different hyperparameter values to Local and Global DEs. The results for self-supervised pre-training are very similar between different datasets. With supervised pre-training, the difference between SSE/StarSSE and Local DE varies for different datasets, probably due to higher variance in optimal hyperparameters. For SUN-397, SSE and StarSSE show very strong results close to the Global DE, while for CIFAR-10, their results are inferior to the Local DE.

## F    Analysis of SSE and StarSSE behavior

In this section, we provide additional plots on the analysis of SSE and StarSSE for the case of supervised pre-training. Figure 12 is analogous to Figure 2 from the main text and demonstrates the behavior of train and test accuracy along line segments between the models from SSE and StarSSE. Figure 13 is analogous to Figure 3 from the main text and demonstrates the quality of individual models from SSE and StarSSE. The general observations are the same as in Sections 5.3 and 5.4. Ultimately, SSE and StarSSE may work in either local or semi-local regimes, but they achieve optimal results in the local one.

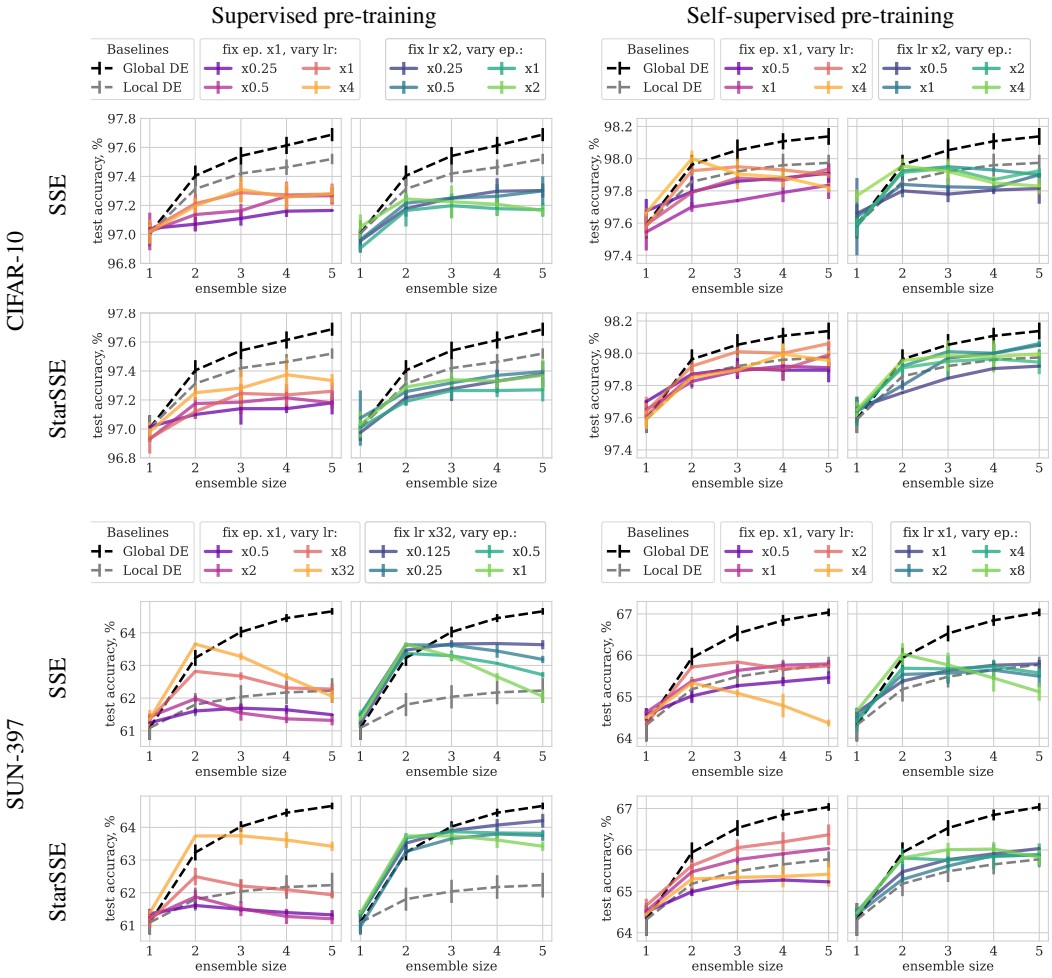

Figure 11: Results of SSEs and StarSSEs of different sizes on CIFAR-10 and SUN-397 for different types of pre-training and varying values of cycle hyperparameters (maximum learning rate and a number of epochs). Local and Global DEs are shown for comparison with gray dotted lines.

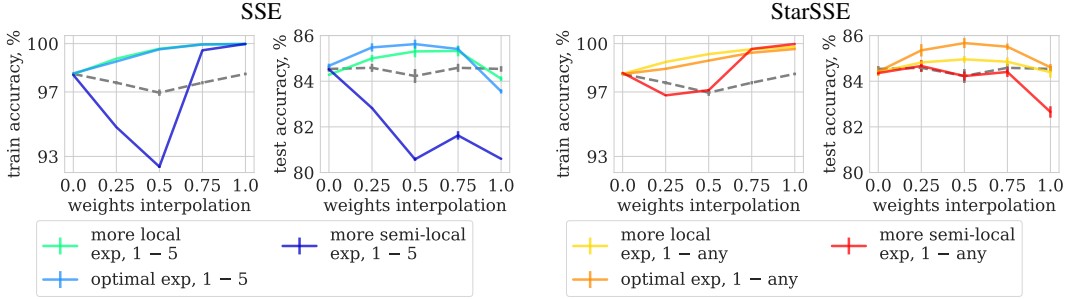

Figure 12: Linear connectivity analysis of Local DEs, SSEs (left plots) and StarSSEs (right plots) on CIFAR-100 with supervised pre-training. We show train and test accuracy along line segments between two random networks in Local DEs, between the first and the last (5-th) network in three differently behaving SSEs, and between the first and any other consequent network in three differently behaving StarSSEs. Hyperparameters for more local and more semi-local experiments are the same for SSE and StarSSE, while hyperparameters for the optimal experiments may differ.

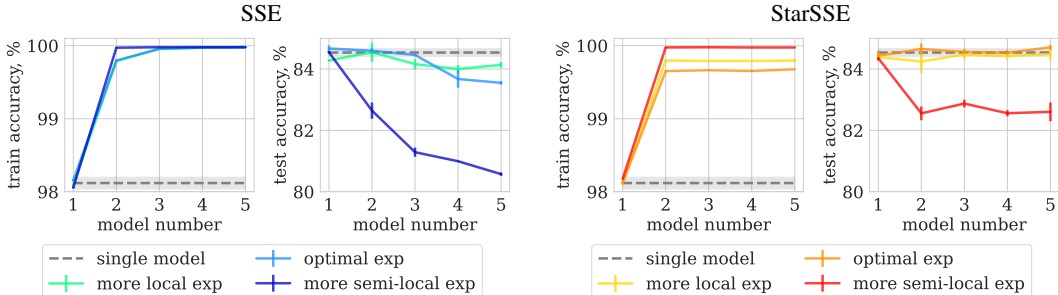

Figure 13: Train and test accuracy of individual models from three differently behaving SSEs (left plots) and StarSSEs (right plots) on CIFAR-100 with supervised pre-training (with a single fine-tuned model for comparison). Hyperparameters for more local and more semi-local experiments are the same for SSE and StarSSE, while hyperparameters for the optimal experiments may differ.

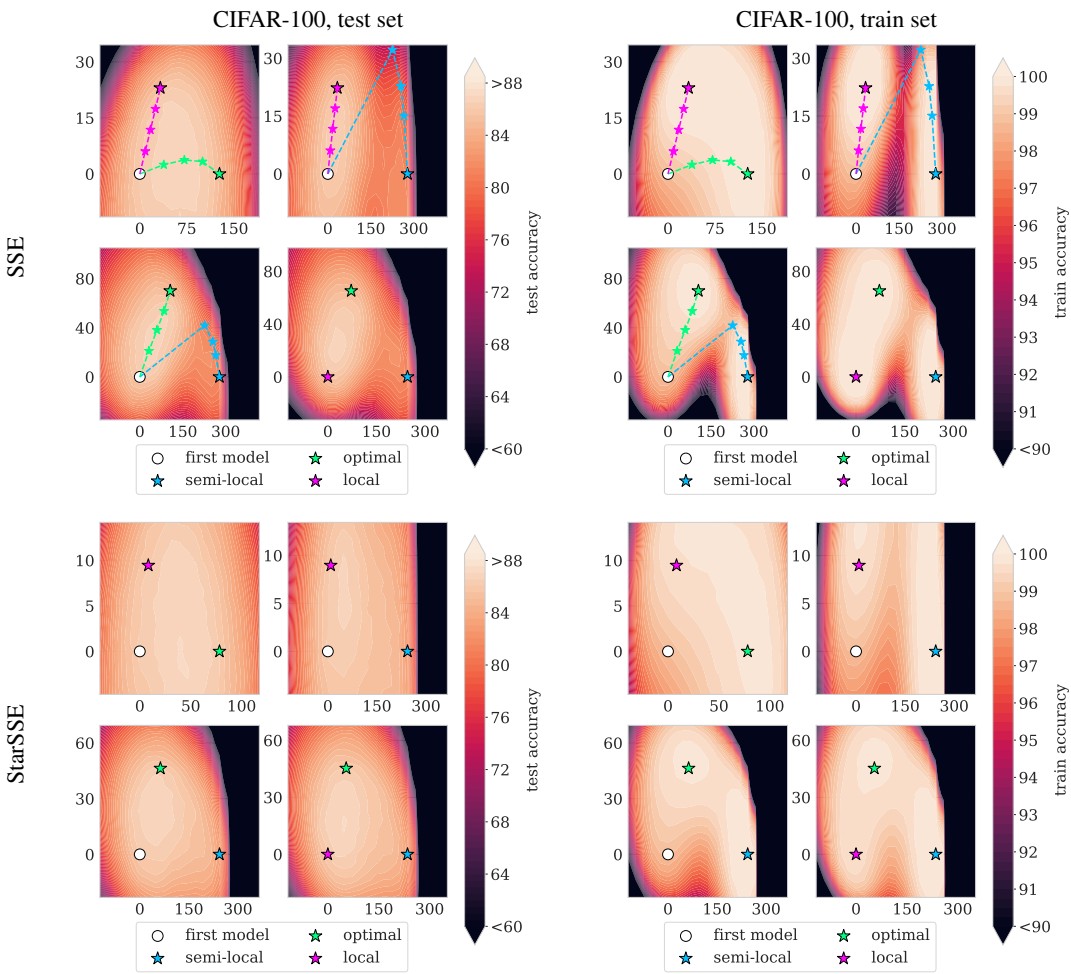

Figure 14: 2D visualization of test and train accuracy around SSE and StarSSE models on the CIFAR-100 dataset with self-supervised pre-training. The markers with black outline correspond to the points generating the projection plane. The axes scale matches the actual distances between the points of the plane in the weight space. The stars without black outline in the SSE plots show the projection of SSE models from previous cycles (note that these are only the projections: the models themselves lay outside of the projection plane and their actual accuracy is different to the one plotted).

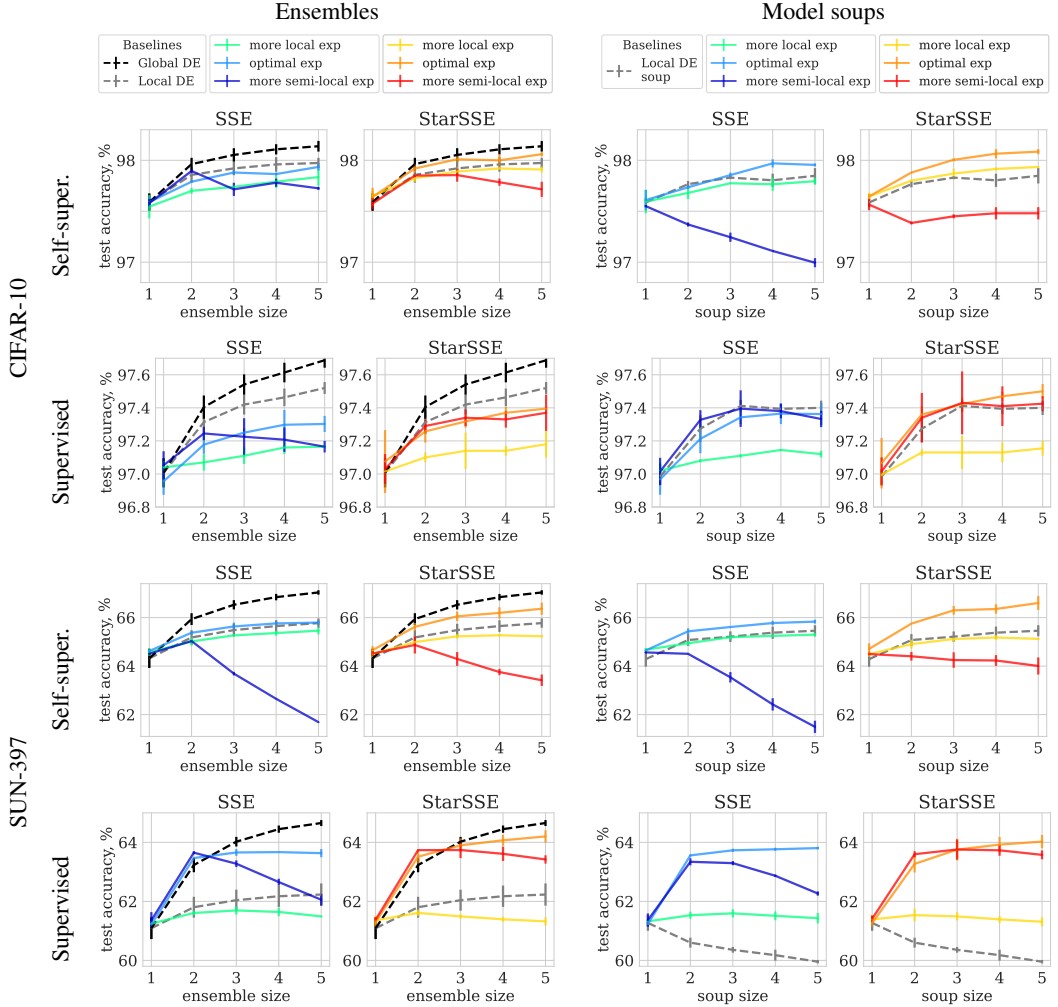

Figure 15: Results of ensembles (left plots) and model soups (right plots) of different sizes on CIFAR-10 and SUN-397 for SSE and StarSSE with supervised and self-supervised pre-training. Hyperparameters for more local and more semi-local experiments are the same for SSE and StarSSE on the same dataset, while hyperparameters for the optimal experiments may differ.

# G  Accuracy behavior on 2D planes in the weight space

In this section, we plot 2-dimensional visualizations of high-accuracy regions around SSE and StarSSE models similar to Garipov et al. (2018). To construct a 2D visualization, one needs to select three points in the weight space and then plot the metric values over the plane generated by these points. Overall, for SSE, we consider four points: the first SSE model (i. e., a model from a Local DE) and three networks from three differently behaving SSE experiments; each network is taken from the last (5-th) SSE cycle. Importantly, the three SSE experiments are all launched starting from the same first model, which is one of the four points considered. We then take all combinations of size three out of these four points and plot separate 2D plots for them. We repeat the same procedure for StarSSE models.

Each combination of three points forms a triangle, which is projected into the image plane. The first model is projected into the point $(0, 0)$. We make the $x$-axis parallel to the side connecting the first and the second models. The $y$-axis is parallel to the altitude of the triangle passing through the third model. The side and the altitude are projected, preserving their original length, so the whole plane is projected without distortion. Finally, we consider a coordinate grid over the projected plane,

update the batch normalization statistics at the grid nodes (we use 10% of the training set for faster processing), and evaluate test and train accuracy at these points.

The resulting 2D visualization plots are shown in Figure 14. In terms of train accuracy, the first and the last local and optimal SSE models are located in the same basin, while the last semi-local SSE model "goes round the corner" — it is almost linearly connected with the last optimal model but we see more significant barriers while connecting it to the first and the last local SSE models. The same observations hold for StarSSE. In terms of test accuracy, the first and the last local and optimal SSE models show similar results, however, the optimal SSE experiment goes much further from the first model and ends up "on the opposite wall" of the high-accuracy region, while the last local SSE model stays "on the same wall" as the first model. This observation holds for StarSSE too. It also matches our model soups experiments, where optimal SSE/StarSSE usually demonstrate substantially stronger results than the local ones.

Semi-local SSE/StartSSE end up very close to the border of the high-accuracy region, both in terms of test and train metrics, in the considered 2-dimensional weight subspaces. When we look at the segment connecting the first and the last semi-local models, we see an abrupt degradation of quality for the models laying beyond the semi-local end. The first and the last local and optimal models are much more stable in this sense. This effect makes us suspect that leaving the pre-train basin makes semi-local SSE/StarSSE converge to sharper minima. Moreover, the track of semi-local SSE experiment shows that the 2-nd SSE cycle moves further from the first model than all subsequent cycles (at least when projected into the image plane). This indicates that even a single cycle with high hyperparameters forces the model to leave the pre-train basin and shows why semi-local StarSSE is not efficient (despite behaving more locally than the semi-local SSE).

## H    Results of soups of SSE and StarSSE models

In this section, we provide additional results of SSE and StarSSE soups on different datasets and with different types of pre-training. Figure 16 is analogous to Figure 4 from the main text, and its first row demonstrates the results on the CIFAR-100 dataset with supervised pre-training. Additionally, Figure 15 shows results for CIFAR-10 and SUN-397 for both supervised and self-supervised pre-training. Similarly to Appendix E, the results for self-supervised pre-training are very similar for all datasets, while for supervised pre-training they vary more. Also, SSE and StarSSE experiments on CIFAR-10 with supervised pre-training become unstable for high hyperparameter values, therefore, the more semi-local experiments here behave similarly to the optimal ones.

## I    OOD

In this section, we show the robustness analysis of SSE and StarSSE with supervised pre-training complementary to Section 7. Figure 16 is analogous to Figure 4 from the main text and compares results on in-distribution CIFAR-100 and out-of-distribution CIFAR-100C data. Results for the supervised case are similar to the self-supervised one: SSE and StarSSE are less stable to the data corruption than Local and Global DEs, however, soups of StarSSE models still show strong results compared to Local DE.

## J    Prediction diversity

In this section, we quantitatively compare the prediction diversity of the networks in SSE, StarSSE, and DE baselines to confirm that the SSE and StarSSE improvement over Local DE is indeed the consequence of higher network diversity. To evaluate the diversity, we use an average normalized prediction difference between model pairs on test data, following Fort et al. (2019) (the higher, the more diverse):

$$diversity = 100 \cdot \mathbb{E}_{m_1 \neq m_2} \frac{\mathbb{E}_{images}[pred_1 \neq pred_2]}{\max(err_1, err_2)},$$

where $m_i$ stands for a model from an ensemble with predictions $pred_i$ and test error level $err_i$. We normalize the prediction difference by the maximum of two errors to pay less attention to the diversity of the models with lower accuracy. Table 5 shows the accuracy of individual models and prediction

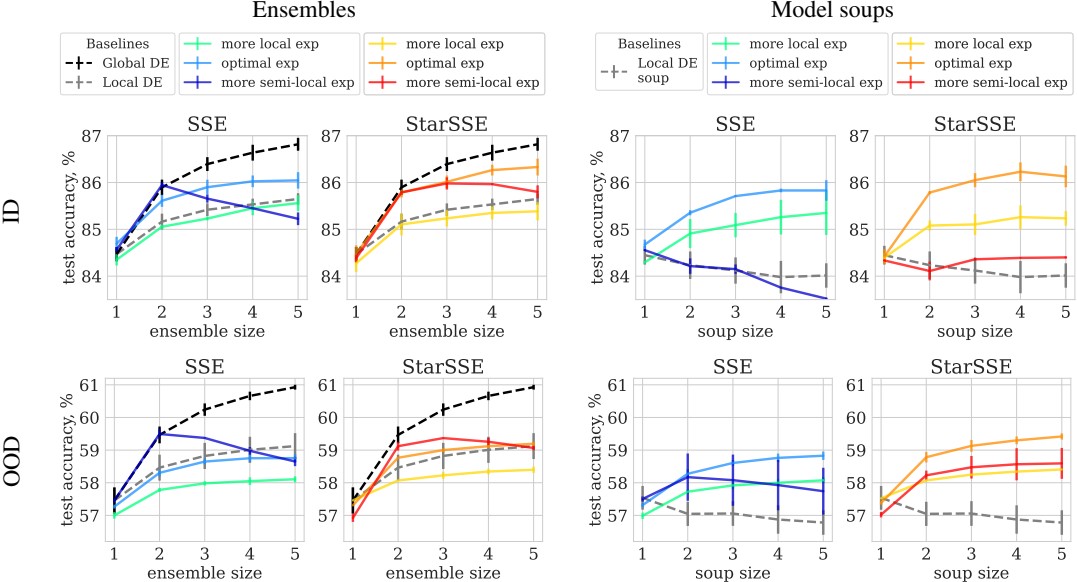

Figure 16: Results of ensembles (left plots) and model soups (right plots) of different sizes on ID (CIFAR-100 test set, top row) and OOD (CIFAR-100C, bottom row) for SSE and StarSSE with supervised pre-training. For OOD, we measure the average accuracy over all possible corruptions and severity values. Standard deviations are calculated over different pre-training checkpoints and/or fine-tuning random seeds.

Table 5: Accuracy of individual models and ensemble diversity for in-distribution (ID, CIFAR-100 test set) and out-of-distribution (OOD, CIFAR-100C) data for supervised (SV) and self-supervised (SSL) pre-training. The OOD results are averaged over different corruptions and values of severity. Standard deviations for both ID and OOD are shown over different pre-training checkpoints and training seeds.

| Pre-train. | Method | | accuracy (single) | | diversity | |
| --- | --- | --- | --- | --- | --- | --- |
| | | | ID | OOD | ID | OOD |
| SV | **Local DE** | | $84.47_{\pm0.15}$ | $57.46_{\pm0.81}$ | $60.21_{\pm1.37}$ | $58.72_{\pm2.33}$ |
| | **Global DE** | | | | $82.58_{\pm1.70}$ | $79.05_{\pm2.32}$ |
| | **SSE** | *more local* | $84.24_{\pm0.23}$ | $56.30_{\pm0.99}$ | $64.01_{\pm8.94}$ | $60.80_{\pm8.32}$ |
| | | *optimal* | $84.18_{\pm0.49}$ | $56.38_{\pm1.27}$ | $70.50_{\pm8.78}$ | $67.79_{\pm9.06}$ |
| | | *more semi-local* | $82.03_{\pm1.45}$ | $54.38_{\pm2.52}$ | $81.35_{\pm12.32}$ | $78.40_{\pm11.63}$ |
| | **StarSSE** | *more local* | $84.35_{\pm0.20}$ | $57.06_{\pm0.69}$ | $55.30_{\pm1.74}$ | $53.62_{\pm2.29}$ |
| | | *optimal* | $84.55_{\pm0.17}$ | $56.77_{\pm1.03}$ | $73.28_{\pm4.42}$ | $69.07_{\pm4.23}$ |
| | | *more semi-local* | $82.95_{\pm0.75}$ | $55.31_{\pm1.80}$ | $82.87_{\pm9.47}$ | $78.63_{\pm9.30}$ |
| SSL | **Local DE** | | $85.96_{\pm0.28}$ | $58.53_{\pm0.82}$ | $66.25_{\pm3.56}$ | $62.75_{\pm3.38}$ |
| | **Global DE** | | | | $80.21_{\pm1.62}$ | $75.93_{\pm2.36}$ |
| | **SSE** | *more local* | $85.87_{\pm0.24}$ | $57.83_{\pm0.85}$ | $47.46_{\pm7.23}$ | $44.61_{\pm7.64}$ |
| | | *optimal* | $85.27_{\pm0.49}$ | $57.56_{\pm1.26}$ | $71.57_{\pm5.94}$ | $66.98_{\pm6.36}$ |
| | | *more semi-local* | $82.52_{\pm1.96}$ | $54.20_{\pm2.94}$ | $80.93_{\pm9.57}$ | $78.68_{\pm10.37}$ |
| | **StarSSE** | *more local* | $85.95_{\pm0.32}$ | $58.13_{\pm0.76}$ | $38.94_{\pm1.57}$ | $36.12_{\pm1.80}$ |
| | | *optimal* | $85.79_{\pm0.28}$ | $57.91_{\pm0.97}$ | $71.51_{\pm2.08}$ | $67.37_{\pm3.12}$ |
| | | *more semi-local* | $83.65_{\pm1.20}$ | $55.48_{\pm2.18}$ | $82.42_{\pm8.59}$ | $79.22_{\pm8.24}$ |

Table 6: Accuracy of individual models and ensembles of size 5 for Local DE, Global DE and optimal StarSSE, trained with and without hyperparameter and augmentation diversification of ensemble members. Self-supervised pre-training.

| Dataset | Diversity | | Local DE | Local DE | Global DE | StarSSE | StarSSE |
| --- | --- | --- | --- | --- | --- | --- | --- |
| | hyp. | aug. | (single) | (ensemble) | (ensemble) | (single) | (ensemble) |
| CIFAR-100 | $-$ | $-$ | $85.96_{\pm0.28}$ | $87.33_{\pm0.21}$ | $88.10_{\pm0.11}$ | $85.79_{\pm0.28}$ | $87.63_{\pm0.12}$ |
| | $-$ | $+$ | $85.95_{\pm0.27}$ | $87.45_{\pm0.15}$ | $88.20_{\pm0.03}$ | $85.91_{\pm0.17}$ | $\mathbf{87.82}_{\pm0.04}$ |
| | $+$ | $-$ | $85.85_{\pm0.28}$ | $\mathbf{87.60}_{\pm0.21}$ | $\mathbf{88.25}_{\pm0.05}$ | $85.69_{\pm0.27}$ | $87.39_{\pm0.06}$ |
| | $+$ | $+$ | $85.24_{\pm0.38}$ | $87.54_{\pm0.14}$ | $88.24_{\pm0.17}$ | $85.81_{\pm0.24}$ | $87.74_{\pm0.08}$ |
| SUN-397 | $-$ | $-$ | $64.32_{\pm0.40}$ | $65.77_{\pm0.19}$ | $67.04_{\pm0.09}$ | $63.98_{\pm0.49}$ | $66.37_{\pm0.25}$ |
| | $-$ | $+$ | $64.25_{\pm0.41}$ | $65.84_{\pm0.29}$ | $66.99_{\pm0.05}$ | $64.23_{\pm0.17}$ | $\mathbf{66.69}_{\pm0.15}$ |
| | $+$ | $-$ | $64.19_{\pm0.52}$ | $66.17_{\pm0.28}$ | $67.33_{\pm0.05}$ | $63.93_{\pm0.40}$ | $66.30_{\pm0.14}$ |
| | $+$ | $+$ | $64.40_{\pm0.46}$ | $\mathbf{66.51}_{\pm0.35}$ | $\mathbf{67.49}_{\pm0.10}$ | $64.30_{\pm0.31}$ | $66.59_{\pm0.06}$ |

diversity on in-distribution CIFAR-100 (ID) and out-of-distribution CIFAR-100C (OOD) data for both types of pre-training.

Despite Local and Global DEs consisting of the models of the same quality level, the latter includes significantly more diverse networks, explaining the quality gap between the two. Generally, when we increase the hyperparameters of SSE/StarSSE, the networks become more diverse, but their ID accuracy decreases. The models of the optimal SSE/StarSSE experiments are much more diverse than the ones of the more local experiments, which makes up for lower ID accuracy when constructing ensembles. Moreover, the high diversity of models in optimal SSE/StarSSE exceeds the one of the Local DE models, allowing optimal StarSSE (and sometimes optimal SSE) to outperform the Local DE on clean data. The models of the more semi-local experiments achieve even higher diversity, though it is not enough to overcome their significant quality degradation.

Our robustness analysis in Section 7 (Figure 4) and Appendix I (Figure 16) shows that optimal SSE and StarSSE are less preferable under the distribution shifts than Local DE. The results in Table 5 confirm that the main reason for this is not the lower diversity but the lower quality of individual models. Generally, SSE and StarSSE models tend to overfit to target data — this is the main reason why semi-local methods do not show positive results in the transfer learning setup. Choosing the optimal hyperparameters for SSE/StarSSE results in low/almost no degradation of individual model quality on ID data (see Figures 3,13 and Table 5). However, overfitting becomes noticeable faster on OOD data, hence, individual SSE/StarSSE model quality degrades more significantly in this case.

## K  Diversifying ensembles

One of the most common techniques to improve ensemble quality is to diversify networks by training them in different ways (varying optimization algorithm, learning rate, weight decay, data augmentation, etc.). This technique is shown to be effective for both Local (Mustafa et al., 2020; Wortsman et al., 2022; Rame et al., 2022) and Global DE (Gontijo-Lopes et al., 2022). In this section, we show that such diversification can also improve StarSSE and thus does not change the results of methods comparison.

We consider ResNet-50 with self-supervised pre-training on CIFAR-100 and SUN-397 datasets and try two types of model diversification:

- *hyperparameter diversification* — instead of training ensemble members with a fixed learning rate, weight decay, and number of epochs, we sample these hyperparameters from the log-uniform distributions on segments around their optimal values:
    - **CIFAR-100:** learning rate — $[0.035, 0.07]$, number of epochs — $[20, 50]$ for Local DE; learning rate — $[0.08, 0.12]$, number of epochs — $[18, 35]$ for StarSSE; weight decay — $[5 \cdot 10^{-5}, 2 \cdot 10^{-4}]$ for both.

Table 7: Accuracy of individual models and ensembles of size 5 for three differently behaving SSEs without regularization, with EWC regularization to the pre-trained model (EWC-PT) and with L2 regularization to the first fine-tuned model (L2-FT) on CIFAR-100 for self-supervised pre-training. StarSSE results are given for comparison.

| | more local | | optimal | | more semi-local | |
| | (single) | (ensemble) | (single) | (ensemble) | (single) | (ensemble) |
|---|---|---|---|---|---|---|
| **SSE** | $85.87_{\pm 0.24}$ | $86.57_{\pm 0.07}$ | $85.27_{\pm 0.49}$ | $87.11_{\pm 0.08}$ | $82.52_{\pm 1.96}$ | $85.60_{\pm 0.10}$ |
| + EWC-PT | $85.83_{\pm 0.21}$ | $86.55_{\pm 0.05}$ | $85.57_{\pm 0.45}$ | $87.22_{\pm 0.05}$ | $82.73_{\pm 1.75}$ | $85.69_{\pm 0.04}$ |
| + L2-FT | $85.94_{\pm 0.16}$ | $86.62_{\pm 0.04}$ | $85.79_{\pm 0.22}$ | $87.26_{\pm 0.05}$ | $85.65_{\pm 0.21}$ | $87.26_{\pm 0.04}$ |
| **StarSSE** | $85.95_{\pm 0.32}$ | $86.41_{\pm 0.30}$ | $85.79_{\pm 0.28}$ | $87.63_{\pm 0.12}$ | $83.65_{\pm 1.20}$ | $86.20_{\pm 0.05}$ |

- - **SUN-397:** learning rate — $[0.1, 0.15]$, number of epochs — $[15, 30]$ for Local DE; learning rate — $[0.1, 0.25]$, number of epochs — $[15, 23]$ for StarSSE; weight decay — $[5 \cdot 10^{-5}, 10^{-4}]$ for both.

- *augmentation diversification* — complementary to fixed data augmentation (random crop and random flip), we use four color augmentations (adjusting brightness, contrast, hue, and saturation) similar to Mustafa et al. (2020), Appendix A.3 (though we needed to change the brightness parameters because of a different parametrization used by torchvision (Paszke et al., 2019)). We sample minimal and maximal limits for the strength of these augmentations before each model training:

  - **brightness:** $s_{\min} \sim Uniform(0.25, 0.75), s_{\max} \sim Uniform(2 \cdot s_{\min}, 4 \cdot s_{\min})$
  - **hue:** $\delta_{\max} \sim Uniform(0.01, 0.2), \delta_{\min} = -\delta_{\max}$
  - **saturation:** $s_{\min} \sim Uniform(0.25, 0.75), s_{\max} \sim Uniform(2 \cdot s_{\min}, 4 \cdot s_{\min})$
  - **contrast:** $s_{\min} \sim Uniform(0.25, 0.75), s_{\max} \sim Uniform(2 \cdot s_{\min}, 4 \cdot s_{\min})$

  During training, each augmentation is applied with 50% probability, and its strength is sampled within the corresponding limits.

For diversified StarSSE, we train the first model with fixed optimal hyperparameters and augmentations and then train all the following models utilizing the diversification techniques.

In Table 6 we show how model diversification influences the average accuracy of individual models and ensembles of size 5 for Local DE, Global DE, and StarSSE. All considered ensemble types benefit from the diversification: while Global and Local DEs improve more from hyperparameter diversification, StarSSE is stronger in combination with augmentation diversification. The latter shows that proper augmentations are very important for StarSSE since longer fine-tuning of StarSSE networks makes them more prone to overfitting. In conclusion, network training diversification can improve the results of all ensemble types and thus can not close the quality gaps between them.

## L   Regularizing SSE to stay in the pre-trained basin

In this section, we experiment with different regularization techniques to prevent SSE from overfitting and losing the advantages of transfer learning. As a first technique, we consider Elastic Weight Consolidation (EWC, Kirkpatrick et al. (2017)), a popular continual learning method, which employs weighted L2 regularization to the pre-trained checkpoint. The weight of each parameter is a diagonal element of the Fisher information matrix (FIM) evaluated on the pre-training task. Hence, EWC not only regularizes the weights to not to go too far from the pre-train checkpoint, but also takes into account the pre-train loss landscape to control the direction of weights' changes. This makes EWC a very similar technique to the informative prior of Shwartz-Ziv et al. (2022), which also utilises the pre-train loss landscape to construct the Gaussian prior on network weights. We consider three values of the regularization coefficient $\lambda_{EWC} \in [0.1, 1, 10]$, choosing the one performing best on the test set $\lambda_{EWC}^* = 10$.

As a second technique, we consider regularization of SSE to the first fine-tuned model to force local behavior. In this case, we use isotropic L2 regularization. The reasons for such a simplification are two-fold: 1) we moved away from the pre-trained checkpoint, making it impossible to evaluate FIM

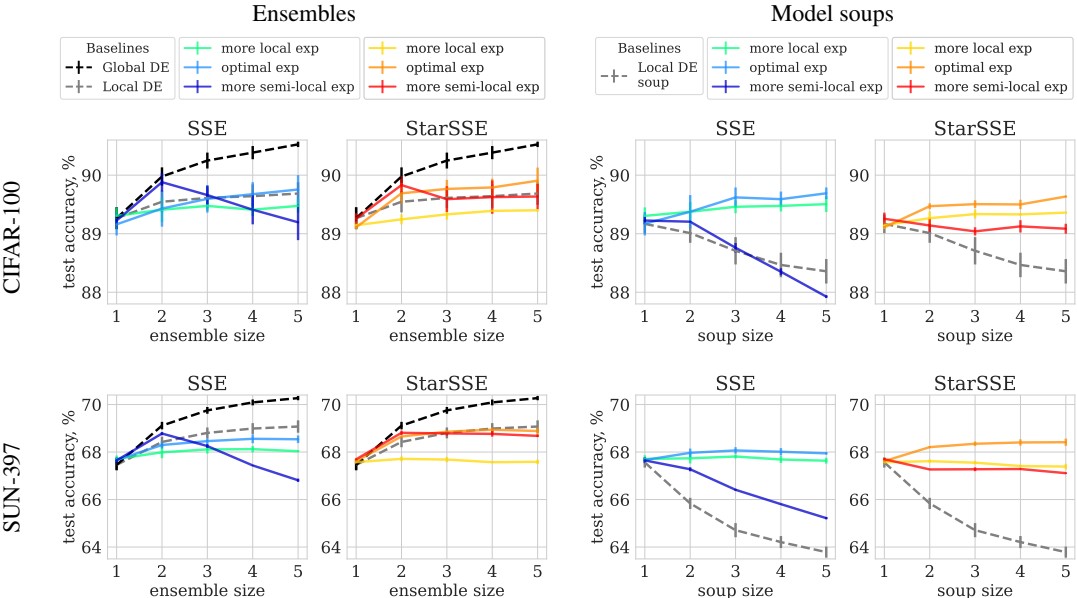

Figure 17: Results of ensembles (left plots) and model soups (right plots) of Swin-T models with supervised pre-training. Three differently behaving SSE and StarSSE experiments on CIFAR-100 and SUN-397 are shown with Local and Global DEs for comparison. Hyperparameters for more local and more semi-local experiments are the same for SSE and StarSSE on the same dataset, while hyperparameters for the optimal experiments may differ.

on the pre-train task without refitting the classifier head, and 2) there is no need to evaluate FIM on the target task, as we already have access to the target loss landscape during the optimization. We consider three values of the regularization coefficient $\lambda_{L2} \in [10^{-5}, 10^{-4}, 10^{-3}]$, choosing the one performing best on the test set $\lambda_{L2}^* = 10^{-4}$. In both cases (EWC to the pre-trained checkpoint and L2 to the first fine-tuned model), the linear head is not regularized.

In Table 7, we consider three differently behaving SSEs and add regularization to each of them, preserving the original hyperparameter values (i. e., the maximum learning rate and the number of epochs for each cycle). Overall, the EWC regularization has little impact on the accuracy of individual models and the ensemble. In contrast, L2 regularization increases both metrics with the most pronounced improvements for the more semi-local experiment, which becomes similar to the optimal one. Generally, both regularization techniques are able to improve the SSE results by promoting a more local behavior and preventing SSE from leaving the pre-train basin, however they can not help to explore the vicinity of the pre-train basin more effectively. Moreover, regularized variants of SSE underperform the optimal StarSSE, making our parallel method the most effective for the local exploration of the pre-train basin.

## M   Additional experiments with Swin-T

In this section, we experiment with Swin-T models pre-trained in a supervised manner. We consider only CIFAR-100 and SUN-397 target tasks because the gap between Local DE and Global DE on CIFAR-10 is too small. For more experimental details see Appendix B. Figure 17 shows the results of ensembles and soups of models from three differently behaving SSE and StarSSE experiments. Generally, we observe the same types of behavior as for ResNet-50. Optimal ensemble variants of SSE and StarSSE show similar results to Local DE, while soups of models from optimal SSE and StarSSE are much more accurate than the ones from Local DE models.

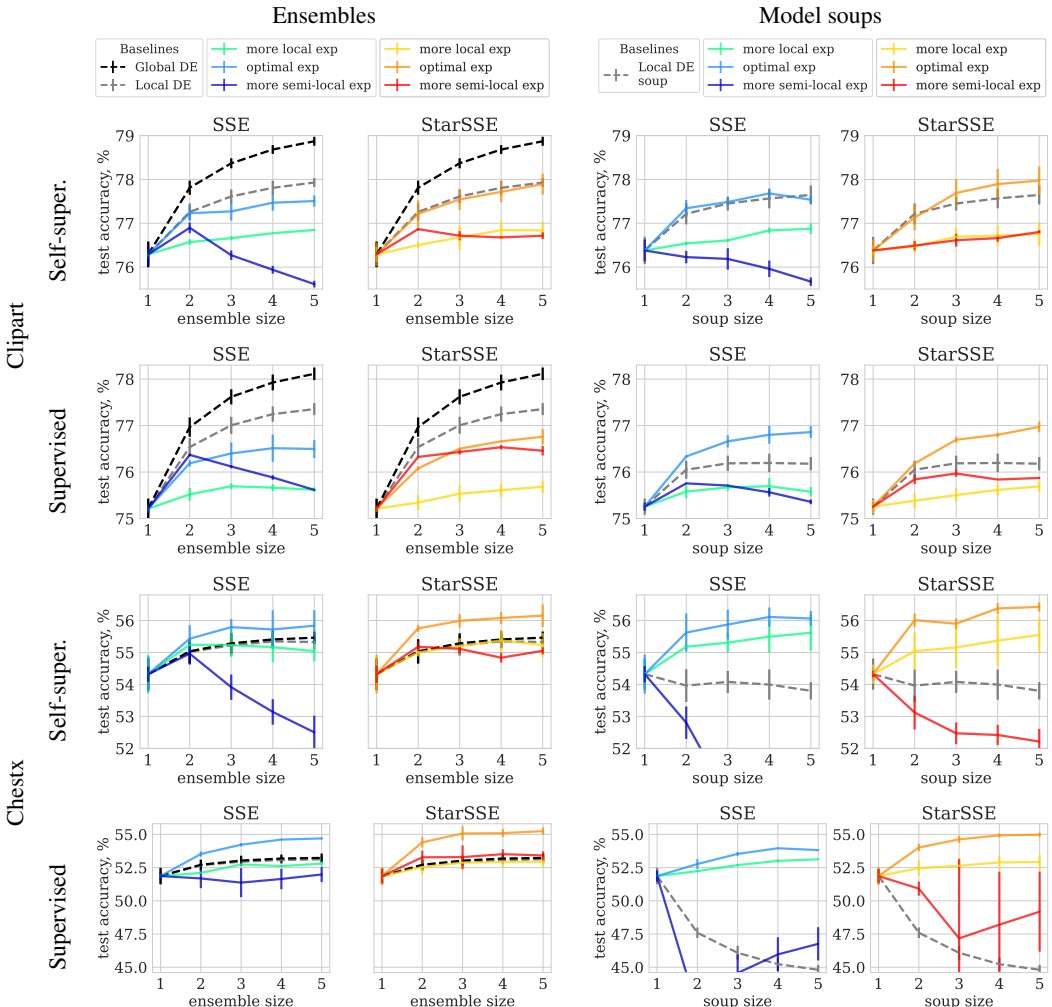

Figure 18: Results of ensembles (left plots) and model soups (right plots) of different sizes on non-natural image classification datasets Clipart and ChestX for SSE and StarSSE with supervised and self-supervised pre-training. Hyperparameters for more local and more semi-local experiments are the same for SSE and StarSSE on the same dataset, while hyperparameters for the optimal experiments may differ.

## N    Additional experiments on non-natural image datasets

In this section, we experiment with two non-natural datasets: ChestX dataset (Wang et al., 2017) containing medical images and a clipart image collection DomainNet Clipart (Peng et al., 2019). We consider the ResNet-50 network with supervised and self-supervised pre-training on ImageNet. Following Ericsson et al. (2021), we drop all multi-label images from ChestX and solve a usual classification problem. For more experimental details, see Appendix B.

Figure 18 shows the results of ensembles and soups of models from three differently behaving SSE and StarSSE experiments. We also provide the quality and diversity of ensemble members in Table 8 for a deeper analysis of StarSSE results. Generally, we observe more extreme results, most likely due to higher discrepancy between source and target tasks.

On Clipart, StarSSE and SSE either show very close or even inferior results compared to the Local DE. Analyzing the quality and diversity of ensemble members, we conclude that StarSSE does not find a balance between the two for this task. In the supervised case, StarSSE finds stronger individual models with lower diversity than the ones of a Local DE. Hence, the ensemble quality of StarSSE

Table 8: Accuracy of individual models and ensemble diversity for experimetns on non-natural image classification datasets with supervised (SV) and self-supervised (SSL) pre-training.

| Pre-train. | Method | Clipart | | Chestx | |
|---|---|---|---|---|---|
| | | acc. (single) | diversity | acc. (single) | diversity |
| SV | **Local DE** | $74.58_{\pm 0.16}$ | $66.74_{\pm 1.65}$ | $51.86_{\pm 0.62}$ | $42.42_{\pm 1.30}$ |
| | **Global DE** | | $75.56_{\pm 1.70}$ | | $43.81_{\pm 1.45}$ |
| | **StarSSE (opt.)** | $75.21_{\pm 0.21}$ | $64.28_{\pm 1.55}$ | $53.93_{\pm 0.47}$ | $46.06_{\pm 4.41}$ |
| SSL | **Local DE** | $75.28_{\pm 0.31}$ | $56.61_{\pm 1.83}$ | $54.32_{\pm 0.49}$ | $38.62_{\pm 1.62}$ |
| | **Global DE** | | $68.46_{\pm 1.46}$ | | $40.05_{\pm 1.76}$ |
| | **StarSSE (opt.)** | $74.58_{\pm 0.16}$ | $67.24_{\pm 1.29}$ | $54.42_{\pm 0.5}$ | $46.03_{\pm 2.70}$ |

is lower, but at the same time, the soup of StarSSE models improves over the Local DE one. In the self-supervised case, models from StarSSE are more diverse but are of lower quality than the Local DE ones. As a result, StarSSE and Local DE show very similar results in this case.

On the contrary, the results on ChestX are very positive: SSE and StarSSE outperform both Local and Global DEs. Interestingly, Local and Global DEs work very similarly on this task. In both supervised and self-supervised cases, StarSSE finds a much more diverse set of models in a more convex part of the loss landscape, hence, resulting in better ensembles and models soups. Additionally, in the supervised case, StarSSE fine-tunes individual models more effectively than Local and Global DEs, improving the results even further.

