# OpenReview forum: "To Stay or Not to Stay in the Pre-train Basin: Insights on Ensembling in Transfer Learning"
_NeurIPS.cc/2023/Conference — NeurIPS 2023 poster_

### Official Review · Reviewer_1VUP · 2023-06-09

**Soundness:** 3 good
**Presentation:** 3 good
**Contribution:** 3 good
**Rating:** 6
**Confidence:** 4

**Summary:**

This paper studies an interesting dilemma -- literature reports that ensembles work better if you end up in different "basins",  but pre-training often confines you to a single "basin". The question is therefore, what should you do? This paper answers that you should start out with a pre-trained model and _stay_ in the basin, but do so using StarSSE -- fine-tune once then fine-tune more times "branching" out from that first fine-tuned model. The solutions can be ensembled and also souped (weight-averaged).

**Strengths:**

This paper tackles an interesting problem. I believe the community will find their results of interest. Some particular strenghts:
- The proposed StarSSE method is well motivated and performs well on the datasets they examine. It seems like a practical modernization of SSE in the era of pre-trained networks.
- I found the result particularly interesting: that if you leave the "basin" of the pre-trained model then you lose the benefits of pre-training.

**Weaknesses:**

There are a few weaknesses that the authors could address to improve the paper:
- The experiments are all very small scale for a paper whose main topic is pre-training. It is understandable to not be able to compare Global DE on larger problems but surely the authors could start out with a pre-trained network when contrasting SSE and StarSSE.
- It seems in Figure 2 / Figure 3 that the runs may be overfitting (much higher train acc then test acc) -- could this be mitigated with standard interventions like label smoothing / data aug / stochastic depth?
- There are terms in the paper which are used frequently but not formalized including "basin". This is OK but then strong claims involving these terms should be avoided (e.g., leaving the basin hurts model performance -- what does this actually mean? that you can no longer interpolate?).

**Questions:**

- How do SSE and StarSSE compare when starting out with a large pre-trained network? For instance if the authors want to stick to image experiments than a pre-trained CLIP model is a good starting point. Here, you can already start out with a good "zero-shot" model.
- How might the results change when using techniques to mitigate overfitting -- it seems like there is overfitting in some of the experiments.

**Limitations:**

I would have like to see a limitations Section which discusses that all experiments are quite small scale.

---

> ### Author Rebuttal · Authors · 2023-08-09
>
> Thank you for the thoughtful review of our work! Please allow us to address your concerns and answer the questions.
>
> **1. Large-scale experiments.**
>
> As you correctly pointed out, in the submission we focused on smaller problems to be able to accurately compare the results with both Local and Global DE baselines. Thank you for the reasonable suggestion to conduct larger experiments dropping the Global DE. We did not manage to make them during the limited rebuttal period, but we will add them in the final revision of the paper.
>
> **2. Overfitting and its mitigation.**
>
> We chose fine-tuning procedures for our experiments based on the standard strategies for specific network architectures and datasets from the literature. For ResNet experiments, we used two types of regularization: weight decay and data augmentation (random crop and random flip), following, e.g., [1]. For Transformers, we additionally used more advanced augmentations (Color Jitter, Auto Augment, Random Erase, Mixup, and Cutmix) and label smoothing following [2]. The test accuracies achieved by our models are close to the results from the literature with similar training setups. A large accuracy gap between train and test metrics is very common for CIFAR datasets (see, e.g., [3]), even if some more advanced regularization techniques are used [4]. Hence, we argue that our experimental setup is adequate.
>
> However, we also think that a deeper investigation of fine-tuning with different data augmentations and regularizations is interesting, especially as a part of the study on the effectiveness of ensemble diversification by varying training hyperparameters for the constituent networks. You can find some preliminary results on this in the answer to question 2 of Reviewer rBpj. We will provide a more broad study in the final revision of the paper.
>
> [1] Grill et. al. Bootstrap your own latent - a new approach to self-supervised learning. NeurIPS 2020.\
> [2] Liu et. al. Swin transformer: Hierarchical vision transformer using shifted windows. ICCV 2021.\
> [3] Garipov et. al. Loss surfaces, mode connectivity, and fast ensembling of DNNs.NeurIPS 2018.\
> [4] https://sebastianraschka.com/blog/2023/data-augmentation-pytorch.html
>
> **3. Formalization of the main terms.**
>
> We give an intuitive definition of the term “basin” in lines 42-43: two models lay in the same basin if there is no high-loss barrier on the linear interpolation between them. The loss barrier is commonly defined as the highest difference between the loss occurring when linearly connecting two points and the linear interpolation of the loss values at each of them [1]. We indeed decided not to formalize our definition of the basin, mostly because we did not want to choose the specific threshold for the loss barrier (it can be exactly 0 or some epsilon>0). Instead, we focused on the loss behavior along the interpolation between the members of different SSE/StarSSE ensembles (Figures 2, 9, and 11) and its comparison to the interpolation of Local DE models. When we say that a model leaves the pre-train basin, we mean that there is a significant train loss barrier between this model and the first fine-tuned model. So, this loss barrier is not only higher than zero but also comparable to or higher than the loss barrier between Local DE models. Empirically we see that when the model leaves the pre-train basin, its test accuracy degrades, and the higher the train loss barrier between the two models, the more severe the test accuracy degradation. We see what you mean by the lack of formalization here and will reformulate this connectivity discussion more accurately in the final revision of the paper.
> Could you please specify which other terms seemed not properly defined to you so we would pay more attention to them in the revision?
>
> [1] Entezari et. al. The role of permutation invariance in linear mode connectivity of neural networks. ICLR 2022.
>
> \
> We hope we adequately addressed most of your concerns given the limited rebuttal period and that you would consider reflecting that in your score. We will also add more large-scale experiments and experiments with other regularizations in the final revision of the paper. If you have any additional questions, we would be happy to answer them during the discussion period =)

---

> > ### Comment · Reviewer_1VUP · 2023-08-12
> > **Thanks for your response.**
> >
> > Thank you for the response, I've increased my confidence. I look forward to seeing the larger scale experiment in the revision.

---

### Official Review · Reviewer_rBpj · 2023-07-03

**Soundness:** 2 fair
**Presentation:** 3 good
**Contribution:** 3 good
**Rating:** 5
**Confidence:** 3

**Summary:**

This paper studies ensembling in the transfer learning setting. This is interesting because pre-training is important for strong predictive performance, yet has been shown to pre-determine the final basin in the loss-landscape after fine-tuning, reducing diversity. The authors explore the behavior of snapshot ensembles(SSE), which use different training iterates as ensemble members, and compare them to Global DE (fine-tune from independent pre-training runs) and Local DE (fine-tune from shared initialization), for both supervised and self-supervised pre-training. The authors find that Global DE is much superior to Local DE, but that Snapshot examples can start to close the gap. They also do some ablations and find that higher learning rates in SSE's increase diversity but decrease overall performance. The authors then propose StarSSE, where a single fine-tuning run seeds the initialization for multiple, parallel fine-tunings. The results of these parallel fine-tuning runs become ensemble members. StarSSE closes the gap between Local DE and Global DE further, and is more amenable to model souping than local DE..


**Strengths:**

Originality:  The authors find that Snapshot Ensembles degrade in performance over subsequent snapshots, a key observation that inspires their "Star SSE". StarSSE is not a particularly large departure from SSE's, but it does seem to address a key limitation,

Quality: The methodology and experimental setup seem sound, so given additional verification (see weaknesses) I would trust the result generalizes.

Clarity :The paper is clear and easy to follow.

Significance: Ensembles and pre-training are both crucial techniques in ML, any attempts to merge the strengths of both in easy-to-implement ways has strong impact.



**Weaknesses:**

Weakness 1: Though results for StarSSE's look promising, only a few datasets are considered. Because different methodologies can be overfit (even unintentionally) to specific datasets, it would be much better to demonstrate efficacy on a larger suite of downstream tasks, as is common (e.g. [1]).

Weakness 2: The baseline alluded to in Line 144 is an important one; "Several recent works (Mustafa et al., 2020; Wortsman et al.,
143 2022; Rame et al., 2022) study another technique for improving the diversity of Local DEs by training
144 networks using different hyperparameters (learning rate, weight decay, data augmentation, etc.)."

The argument that it cannot close the gap between Local and Global DE is not convincing; the given constraints are that you have target data and a pretrained model, and the goal is the best possible performance on target test data. The "diversivied local DE" is a logical choice for this setup, and it's important to charactherize its performance compared to StarSSE.

[1] How Well Do Self-Supervised Models Transfer? - Ericsson et. al., CVPR 2021.

**Questions:**

Please see Weaknesses.

Q1(Weakness 1): How does StarSSE perform on additional downstream datasets, especially non-natural ones?

Q2 (Weakness 2): How does StarSSE compared to "diversifed Local DE's" (line 144)?

With adequate addressing of these questions, I'd be willing to bump my score.

**Limitations:**

The authors have appropriate addressed limitations.

---

> ### Author Rebuttal · Authors · 2023-08-09
>
> Thank you for the thoughtful review of our work! Please allow us to address your concerns and answer the questions.
>
> **1. Experiments on additional downstream tasks.**
>
> We started our project using the same standard benchmark suite (Aircraft, Caltech101, Cars, CIFARs, etc.) used in the work you mentioned [1]. However, due to the limited computational resources, we decided to focus on CIFARs and SUN397 since the gap between Local and Global DEs was the most prominent for these datasets.
>
> We fully agree that experiments on more non-natural datasets, which you suggested, would be an interesting addition to the paper. In the table below we provide the results of such experiments on two datasets: ChestX dataset [2] containing medical images and a clipart image collection DomainNet Clipart [3]. Following [1], we drop all multi-label images from ChestX and solve a usual classification problem. We compare the accuracy of ensembles and model soups of 5 networks with SSL pre-training.
> |||Local DE|Global DE|StarSSE (opt)|
> |---|---|---|---|---|
> |Clipart|Ensemble|77.73 ± 0.23|78.57 ± 0.17|77.69 ± 0.09|
> ||Soup|77.76 ± 0.22|-|77.79 ± 0.11|
> |ChestX|Ensemble|55.33 ± 0.22|55.46 ± 0.19|56.15 ± 0.35|
> ||Soup|53.80 ± 0.27|-| 56.42 ± 0.12|
>
> On Clipart, StarSSE and Local DE results are very close both for ensembles and model soups. This shows that in some cases Local DE models converge to the same convex basin (hence, the high soup quality), and in this case, StarSSE and Local DE explore almost the same area in the loss landscape. The situation with ChestX is quite surprising: StarSSE ensemble outperforms both Local and Global DEs. We analyzed the constituent models and found that the high performance of StarSSE on ChestX originate not from the stronger individual models (individual StarSSE models achieve 54.42 ± 0.50, which is very close to the individual Local/Global DE models with 54.32 ± 0.49) but from the higher diversity (46.00 ± 2.6 for StarSSE v.s. 38.13 ± 1.72 for Local DE and 40.32 ± 1.93 for Global DE). See the answer to question 2 of Reviewer XbMH for the definition of the used diversity metric. We will continue our investigation of the reasons for such positive behavior.
>
> We will include a broader version of the results on non-natural datasets in the final revision of the paper. To further support the paper's conclusions, we will also add large-scale experiments following the suggestion of Reviewer 1VUP.
>
> [1] Ericsson et. al. How Well Do Self-Supervised Models Transfer? CVPR 2021.\
> [2] Wang et. al. ChestXray8: Hospital-scale chest X-ray database and benchmarks on weakly-supervised classification and localization of common thorax diseases. CVPR, 2017.\
> [3] Peng et. al. Moment Matching for Multi-Source Domain Adaptation. ICCV 2019.
>
> **2. Improving network diversity by varying training hyperparameters.**
>
> Varying training hyperparameters of member networks is indeed a useful technique that should be used in practice to obtain the best possible ensembles. What we meant in the paper is that such diversification is known to be effective for both Local and Global DE and can not close the gap between them. However, we agree that experiments on diversified ensembles would be an interesting addition to the paper. We decided not only to compare "diversified Local DE" and StarSSE but also to check whether StarSSE can benefit from such diversification. Below we provide results for ensembles of size 5 on CIFAR-100 with SSL pre-training. We try two types of model diversification:
> * hyperparameter diversification — instead of training ensemble members with a fixed learning rate, weight decay, and number of epochs, we sample these hyperparameters from segments around the optimal values,
> * augmentation diversification — complementary to fixed data augmentation (random crop and random flip), we use four color augmentations (adjusting brightness, contrast, hue, and saturation) similar to [1], Appendix A.3. We sample minimal and maximal limits for the strength of these augmentations before each model training. During training, each augmentation is applied with 50% probability, and its strength is sampled within the corresponding limits.
>
> For diversified StarSSE, we trained the first model with fixed optimal hyperparameters and augmentations as in the paper and then trained all the following models utilizing the diversification techniques.
> ||Local DE|Global DE|StarSSE (opt)|
> |---|---|---|---|
> |No diversification|87.33 ± 0.21|88.10 ± 0.11|87.63 ± 0.12|
> |Hyperparameter diversification|87.60 ± 0.21|88.25 ± 0.05|87.39 ± 0.06|
> |Augmentation diversification|87.45 ± 0.15|88.20 ± 0.03|87.82 ± 0.04|
>
> The results show that all the methods can benefit from the diversification of network training. While Global and Local DEs improve more from hyperparameter diversification, StarSSE is stronger in combination with augmentation diversification. We will include more thorough experiments with diversified ensembles in the final revision of the paper.
>
> [1] Mustafa et. al. Deep Ensembles for Low-Data Transfer Learning. 2020.
>
> \
> We hope we adequately addressed most of your concerns given the limited rebuttal period and that you would consider reflecting that in your score. We will also add more detailed experiments in the final revision of the paper. If you have any additional questions, we would be happy to answer them during the discussion period =)

---

> > ### Comment · Reviewer_rBpj · 2023-08-10
> > **Acknowledgment**
> >
> > Thank you to the authors for the rebuttal, which answered my questions and concerns. I've increased my score to reflect my increased confidence in the findings of the paper.

---

### Official Review · Reviewer_a9o6 · 2023-07-06

**Soundness:** 3 good
**Presentation:** 2 fair
**Contribution:** 2 fair
**Rating:** 6
**Confidence:** 3

**Summary:**

This paper focuses on ensembling in transfer learning scenarios, i.e., constructing ensembles starting from only one pre-trained checkpoint. Empirical results employing Snapshot Ensembles (SSE; Huang et al., 2017) demonstrate that leaving a pre-trained basin can degrade the quality of the ensemble. Based on this, the authors present a variant of SSE, StarSSE, which forces fine-tuned ensemble members to stay in the pre-trained basin.

**Strengths:**

The empirical analysis conducted using Local DE, Global DE, and SSE is convincing and supports the main argument that staying in the pre-trained basin is beneficial for constructing ensembles in transfer learning scenarios. While earlier studies (Neyshabur et al., 2020; Wortsman et al., 2022) have already demonstrated the tendency of fine-tuned solutions to stay within the same pre-trained basin, this study goes a step further to argue that diverging from the pre-trained basin to enhance ensemble diversity does not result in improvements in ensemble performance. In summary, the paper provides insights into the current trend of favoring the fine-tuning of large-scale models over training them from scratch.

**Weaknesses:**

The proposed StarSSE is equivalent to the Local DE using a fine-tuned checkpoint instead of a pre-trained checkpoint as initialization. Although the experimental analysis was conducted using the SSE, I am not sure whether we can call the proposed algorithm as a "snapshot." In contrast to the original snapshot approach, which employs a single continuous SGD trajectory with cyclical learning rates to explore loss surfaces, the proposed StarSSE method collects samples in the vicinity of the first fine-tuned solution through multiple short SGD runs.

Moreover, concerning this aspect, it is strongly recommended to verify the comparative performance of the proposed StarSSE method against other "local" techniques which collect samples near the fine-tuned solution. Specifically, the following methods can be considered within the transfer learning context:
* Constructing a Laplace approximation at the same fine-tuned solution and ensembling the obtained samples.
* Constructing a SWA-Gaussian approximation centered around the fine-tuned solution (which would be the SWA solution in this case) and ensembling the collected samples.


**Questions:**

1. Does the diversity observed among the ensemble members in StarSSE originate solely from distinct SGD noise sources, such as random mini-batch order and data augmentation techniques?
2. There are alternative ways to regularize fine-tuned solutions to stay in the pre-trained basin, e.g., Xuhong et al. (2018) and Shwartz-Ziv et al. (2022). It would be nice to compare the initialization with the starting point (StarSSE) and the regularization towards the starting point (e.g., L2 regularization, like Xuhong et al., 2018).
3. Having visual representations of two-dimensional loss surfaces would be advantageous in providing supplementary evidence for the empirical analysis. According to the paper, it is expected that configurations described as "more semi-local" would deviate from the basin, while "optimal" and "more local" setups would stay within the basin.

**Limitations:**

Appendix A addresses the limitations.

---

> ### Author Rebuttal · Authors · 2023-08-09
>
> Thank you for the thoughtful review of our work! Please allow us to address your concerns and answer the questions.
>
> **W1. StarSSE name and connection to Local DE.**
>
> Your assessment of the conceptual closeness of StarSSE and Local DE is absolutely correct, and we discuss it in the paper (lines 295-304). We had the same doubts as you about the name but agreed on StarSSE because it was originally motivated as a modification of SSE in our project. Generally, we are open to changing the name to, e.g., Star Ensemble or Star DE, especially if other reviewers also find the name StarSSE a bit misleading. We would also be happy to hear your suggestions =)
>
> **W2. Comparison of StarSSE and other local ensemble techniques.**
>
> We initially focused on the cyclical methods (SSE, FGE, cSGLD) because they are the only semi-local methods and, at the same time, they show strong performance as local methods in the standard setup (see results of FGE in [1]). However, we agree that additional comparisons with other local methods would benefit the StarSSE part of the paper. Below we provide comparisons with KFAC-Laplace [2] and SPRO [3], both of which sample models from a local area near the usually trained network. Since SWAG is based on the SWA solution, its benefits are not only due to ensembling but also due to obtaining a better base model. Hence, for a fair comparison with SWAG, our method and baselines should be modified by incorporating SWA at the end of the training of each model. We were not able to conduct the experiments with SWA due to the time limitations of the rebuttal period, however, we will include them in the final revision of the paper.
>
> The comparison of different 5-network ensembles on CIFAR-100 with SSL pre-training is shown in the table below (we also measure the diversity of networks in ensembles, see the answer to question 2 of Reviewer XbMH for the definition of the used diversity metric). We used grid search to choose an optimal sampling scale (1.0) for KFAC-Laplace and the optimal values of the learning rate (0.05) and training epochs (20) for each new vertex, as well as the regularization coefficient (1e-5) for SPRO.
> ||Local DE|StarSSE (opt)|KFAC-Laplace|SPRO|
> |---|---|---|---|---|
> |Test acc|87.33±0.21|87.63 ± 0.12|86.01 ± 0.02|86.46 ± 0.12|
> |Diversity|66.08 ± 3.42|71.46 ± 2.01|35.63 ± 2.24|42.43 ± 18.27|
>
> KFAC-Laplace and SPRO results are inferior to StarSSE and, moreover, to Local DE, meaning these methods are extremely local. The results are consistent with the results for the standard setup from [1].
>
> [1] Ashukha et. al. Pitfalls of in-domain uncertainty estimation and ensembling in deep learning. ICLR 2020.\
> [2] Ritter et. al. A Scalable Laplace Approximation for Neural Networks. ICLR 2018.\
> [3] Benton et. al. Loss surface simplexes for mode connecting volumes and fast ensembling. ICML 2021.
>
> **Q1. The origin of diversity among the ensemble members in StarSSE.**
>
> Yes, the only origin of diversity is random mini-batch order and data augmentation. However, due to the separation between fine-tuning and exploration, we can utilize higher hyperparameter values (learning rate and number of epochs) in StarSSE, which results in higher diversity. We also provide the additional diversity analysis answering question 2 of Reviewer XbMH, which you may find interesting in connection to this question.
>
> **Q2. Other ways to regularize SSE solutions to stay in the pre-trained basin.**
>
> Inspired by [1], we tried Elastic Weight Consolidation (EWC) towards the pre-trained solution (L2 regularization, reweighted using the approximation of the Fisher matrix on the source task) after the submission. Our motivation was to allow SSE to account for the source loss landscape shape when exploring the target loss landscape. We obtained a slight improvement of the optimal results: for ensembles of size 5 on CIFAR-100 with SSL pre-training, we get 87.22 ± 0.05 for SSE + EWC v.s. 87.11 ± 0.08 for SSE (for comparison, for StarSSE we get 87.63 ± 0.12). However, the optimal experiment still behaved as a local method, and EWC did not help SSE explore different target loss basins without losing the transfer benefits, most likely due to its Gaussian prior shape.
>
> Following your suggestion, we also tried L2 regularization towards the first fine-tuned network. It improves the optimal results too: 87.26 ± 0.05 in the same setup as above, which is still inferior to the StarSSE results. Hence, initialization with the starting point is more useful than the regularization towards the starting point for making SSE applicable in the transfer learning setup.
>
> We will conduct a more thorough comparison of the discussed regularization techniques and include it in the final revision of the paper.
>
> [1] Shwartz-Ziv et. al. Pre-train your loss: Easy Bayesian transfer learning with informative priors. NeurIPS 2022.
>
> **Q3. Two-dimensional loss surface visualizations.**
>
> In our connectivity analysis in Figures 2, 9, and 11, we compared three types of SSE/StarSSE behaviors, therefore, we decided to look only at one-dimensional visualizations since we can not look at four points (the first fine-tuned model and three final models) on one two-dimensional plain. However, we agree that looking at two-dimensional visualizations could be interesting. For example, a plot for the optimal and semi-local experiment pair may show how SSE with high hyperparameter values “go round the corner” in the train loss landscape and, at the same time, leave the high-quality zone in the test loss landscape. Would you like to suggest any other two-dimensional plains which we should check?
>
> \
> We hope we adequately addressed most of your concerns and that you would consider reflecting that in your score. We will also add more detailed experiments on local methods and regularizations in the final revision of the paper. We would be happy to hear your suggestions regarding W1 and Q3 and answer any additional questions during the discussion period =)

---

> > ### Comment · Reviewer_a9o6 · 2023-08-12
> >
> > Thank you for the authors' efforts.
> >
> > > __Q3. Two-dimensional loss surface visualizations.__
> > >
> > > In our connectivity analysis in Figures 2, 9, and 11, we compared three types of SSE/StarSSE behaviors, therefore, we decided to look only at one-dimensional visualizations since we can not look at four points (the first fine-tuned model and three final models) on one two-dimensional plain. However, we agree that looking at two-dimensional visualizations could be interesting. For example, a plot for the optimal and semi-local experiment pair may show how SSE with high hyperparameter values “go round the corner” in the train loss landscape and, at the same time, leave the high-quality zone in the test loss landscape. Would you like to suggest any other two-dimensional plains which we should check?
> >
> > As the authors have pointed out, visually representing four points within a single plot is not trivial since a two-dimensional plane is defined by three non-collinear points. To address this, an alternative strategy is to draw four separate plots, each depicting a loss surface containing three of the given points. These visualizations would provide readers with a more intuitive understanding of the underlying processes.

---

> > ### Comment · Reviewer_a9o6 · 2023-08-12
> >
> > > __W1. StarSSE name and connection to Local DE.__
> > >
> > > Your assessment of the conceptual closeness of StarSSE and Local DE is absolutely correct, and we discuss it in the paper (lines 295-304). We had the same doubts as you about the name but agreed on StarSSE because it was originally motivated as a modification of SSE in our project. Generally, we are open to changing the name to, e.g., Star Ensemble or Star DE, especially if other reviewers also find the name StarSSE a bit misleading. We would also be happy to hear your suggestions =)
> >
> > Even though the StarSSE algorithm no longer precisely aligns with the definition of a "snapshot," I believe that the main contribution of this paper is rooted in the experimental analysis involving "snapshot" ensembles (SSE). Considering this perspective, it would be more beneficial to stick with the name "SSE", even if the proposed algorithm departs from the original concept of "snapshot." Thank you for sharing a behind-the-scenes :)

---

> > > ### Author Response · Authors · 2023-08-16
> > >
> > > **W1. StarSSE name and connection to Local DE.**
> > >
> > > Since you agreed and other reviewers did not raise concerns regarding the name, we will keep it as is.
> > >
> > > **Q3. Two-dimensional loss surface visualizations.**
> > >
> > > Thank you again for the suggestion! We have plotted four two-dimensional visualizations you suggested and found them very useful to illustrate our conclusions from the paper. Moreover, two dimensions and a wider field of view (we looked at the area substantially larger than the triangle of the three base points in the same manner as was done, e.g., in Figure 1 in [1]) allowed us to make some interesting observations that were not obvious from our one-dimensional plots:
> > > * In terms of train accuracy, the first and final local and optimal SSE models are located in the same basin, while the final semi-local SSE model “goes round the corner” as we expected - it is almost linearly connected with the final optimal model, but we see more significant barriers while connecting it to the first and final local SSE models. The same observations hold for StarSSE.
> > > * In terms of test accuracy, the first and the final local and optimal SSE models show similar results, however, the optimal SSE experiment goes much further from the first model and ends up “on the opposite wall” of the high-accuracy region, while the final local SSE model stays “on the same wall” as a first model. This observation holds for StarSSE too. It also matches our model soups experiments, where optimal SSE/StarSSE usually demonstrate substantially stronger results than the local ones.
> > > * Semi-local SSE/StartSSE end up very close to the border of the high-accuracy region, both in terms of test and train metrics, in the considered two-dimensional weight subspaces. When we look at the segment connecting the first and the final semi-local model, we see an abrupt degradation to the random level quality for the models laying beyond the semi-local end (models with weights $(1-\alpha)\theta_1+\alpha\theta_{sl}$, where $\theta_1$ and $\theta_{sl}$ are the weights of first and the final semi-local models respectively, show random results for $\alpha>1.2$). The first and the final local and optimal models are much more stable in this sense. This effect makes us suspect that leaving the pre-train basin makes semi-local SSE/StarSSE converge to sharper minima.
> > >
> > > We will add these plots in the final revision of the paper (if we understand correctly, we can not add plots to the answers at the discussion stage).
> > >
> > > [1] Garipov et.al. Loss Surfaces, Mode Connectivity, and Fast Ensembling of DNNs. NeurIPS 2018.

---

> > > > ### Comment · Reviewer_a9o6 · 2023-08-17
> > > >
> > > > I am pleased to know that the paper has gained greater solidity. With the expectation that the final manuscript will comprehensively address all the concerns raised by the reviewers, I have raised my score to 6. Thank you again for the further clarification and all the efforts from the authors.

---

### Official Review · Reviewer_XbMH · 2023-07-07

**Soundness:** 2 fair
**Presentation:** 3 good
**Contribution:** 2 fair
**Rating:** 6
**Confidence:** 4

**Summary:**

The authors consider techniques to faciliatate more efficient ensembling when in the transfer learning context.They propose a novel cyclical learning rate method, StarSSE, and compare it to standard methods like Snapshot Ensembles (SSE). They demonstrate that leaving the pretraining basin leads to worse performance when forming ensembles from a pretrained checkpoint, and that cyclical learning rate methods can help to close the gap between purely local and global ensembling methods. Finally, they discuss their methods when forming model soups, and in the context of robusness.

**Strengths:**

- The authors provide a thoughtful experimental analysis of different fine-tuning setups (different models, datasets, pre-training, learning schedules, hyperparameters, fine tuning paradigms) that address an important question: the efficacy of ensembling methods when fine tuning. Their insights into the correspondence between good ensemble members and staying within the pre-training basin are interesting.
- The authors provide a descriptive overview of a wide variety of ensembling methods, and provide categorizations into groups based on their coverage of the loss landscape, as are relevant to their work.

**Weaknesses:**

- My biggest concern is that at the moment, it is difficult to quantitatively evaluate the claim that StarSSE ensembles perform better than SSE ensembles. Do StarSSE ensembles provide a significant improvement over SSE ensembles in the different ensembling conditions that you study? It is difficult for me to determine this from the plots provided in Figure 2, 4 and Appx. D. Could you provide a table of results, perhaps only for the optimal hyperparameter settings at a fixed model number?
- The results on model soups and robustness appear slightly disjoint from the rest of the paper. A more thorough description of related work around robustness and models soups would be useful. It is also unclear why SSE and StarSSE underperform Local Deep Ensembles on OOD data- as robustness is a major reason for practitioners to use deep ensembles, at least understanding why this loss in performance occurs is important.

**Questions:**

- The original conception of Snapshot Ensembles relies upon the model converging to and escaping from local minima. Although the algorithm has been repurposed here to better explore the vicinity of a single local minimum, I am curious how the training loss fluctuates across multiple cycles of SSE as a result.
- Throughout the main text, the diversity of ensembles created through StarSSE is discussed as a benefit. However, since StarSSE models rely only upon the randomness of SGD based training to generate diversity I would not expect them to be necessarily more diverse than SSE, or even Local DE (as would be supported by OOD results). Can you provide some metric of the predictive diversity in Local DE, SSE, and StarSSE ensembles?

**Limitations:**

As the authors acknowledge, the choice of hyperparameter tuning is a potential limitation of the applicability of their study.

---

> ### Author Rebuttal · Authors · 2023-08-09
>
> Thank you for the thoughtful review of our work! Please allow us to address your concerns and answer the questions.
>
> **W1. Unclear quantitative comparison of SSE and StarSSE ensembles.**
>
> Thank you for pointing out that flaw in our presentation. In all the experiments StarSSE ensembles showed significant improvement over SSE. Below we provide a table of results for ensembles of 5 networks trained with optimal hyperparameter settings (see Table 3 in the Supplementary for the specific values). The results for Local and Global DEs of the same size are given in Table 1 in the paper. We will include results in numerical form in addition to figures in the final revision of the paper.
> ||SV SSE|SV StarSSE|SSL SSE|SSL StarSSE|
> |---|---|---|---|---|
> |CIFAR-10|97.30 ± 0.05|97.39 ± 0.01|97.94 ± 0.04|98.06 ± 0.02|
> |CIFAR-100|86.04 ± 0.18|86.30 ± 0.18|87.11 ± 0.06|87.63 ± 0.12|
> |SUN-397|63.63 ± 0.14|64.2 ± 0.21|65.79 ± 0.10|66.37 ± 0.25|
>
> **W2 (part 1). Relevance of model soups and robustness sections.**
>
> We briefly described our reasons for including these parts at the beginning of the corresponding sections. Since model soups are a more practical alternative to local ensembles and StarSSE constructs a high-quality local ensemble, we were interested to see how StarSSE models behave when averaged. And we obtained unexpected results: even though StarSSE models are more diverse, they lay in a much more convex region of the loss landscape. The goal of the robustness analysis was to compare StarSSE and Local DE in more challenging and practical conditions. The results showed an interesting limitation of StarSSE: it is less robust to the test data corruption as an ensemble method, however, averaging StarSSE models is still an effective technique even in this setup. We will try to explain our motivation for these sections more thoroughly in the final revision of the paper and discuss more related works following your suggestion.
>
> **Q1. Training dynamics of local SSE.**
>
> SSE in our experiments shows very similar behavior to the standard setup: train and test loss increase at the beginning of each cycle and then converge at the end of a cycle. Depending on the hyperparameters, the magnitude of the loss increase changes (the higher the learning rate, the higher the loss increase at the beginning of a cycle) as well as the locality of the method (the higher the learning rate and number of epochs, the further the next checkpoint ends up). Varying SSE hyperparameters similarly influence the locality of the method in standard and transfer learning setups. The only difference is that in standard setup training can still converge to a good solution even after leaving the previous local minima, while in a transfer learning setup leaving the pre-train basin results in overfitting and losing the transfer benefits. Hence, when we train SSE with high hyperparameter values in a transfer learning setup, we obtain low train loss at the end of all cycles, but the test loss degrades with each cycle.
>
> We would also like to mention that FGE can be viewed as a local variant of SSE with a different learning rate schedule. Hence, cyclical methods were also used to explore the vicinity of a single local minimum in the standard training setup in the literature. The authors of FGE see the cyclical behavior of the test accuracy but with low magnitude.
>
> **Q2 and W2 (part 2). Diversity of StarSSE models and OOD results.**
>
> Thank you for this interesting question. We conducted some additional analysis of our results, and we will definitely include this analysis in the final revision of the paper.
> Firstly, we computed the predictive diversity metric on both ID and OOD data. We chose an average normalized prediction difference between model pairs based on [1] (the higher, the more diverse):
> $$
> 100 \times \mathbb{E}_{m_1 \ne m_2} \frac{\mathbb{E} [pred_1 \ne pred_2]}{\max (err_1, err_2)},
> $$
>
> where $m_i$ stands for a model from an ensemble with predictions $pred_i$ and test error level $err_i$, the inner expectation is taken over test images.
>
> The results for CIFAR-100 with self-supervised pre-training are shown in the table below. Optimal variants of SSE and StarSSE are indeed more diverse than Local DE on both ID and OOD data. Even though the only source of diversity in SSE and StarSSE is the randomness of SGD-based training, these methods benefit more from this source than Local DE. Due to the separation of fine-tuning and exploration, SSE and StarSSE can effectively use higher hyperparameter values in their cycles, which improves diversity.
>
> |||Local DE|Global DE|SSE (opt)|StarSSE (opt)|
> |---|---|---|---|---|---|
> |Diversity|ID|66.08 ± 3.42|80.03 ± 1.55|71.19 ± 5.71|71.46 ± 2.01|
> ||OOD|62.68 ± 3.41|75.89 ± 2.37|66.91 ± 6.33|67.39 ± 3.09|
> |Test acc (ind model)|ID|85.96 ± 0.28|85.96 ± 0.28|85.27 ± 0.49|85.79 ± 0.28|
> ||OOD|58.53 ± 0.82|58.53 ± 0.82|57.56 ± 1.26|57.91 ± 0.97|
>
> So, why are the results of StarSSE and SSE less preferable to the Local DE on OOD data? The main reason is not the lower diversity but the lower quality of individual models. Generally, we see in our experiments that models tend to overfit the target data — this is the main reason why semi-local methods do not show positive results in the transfer learning setup. Choosing the optimal hyperparameters for SSE/StarSSE resulted in low/almost no degradation of individual model quality on ID data (see Figure 3 and the table above). However, overfitting becomes noticeable faster on OOD data, hence, individual SSE/StarSSE model quality degrades more significantly in this case. Nevertheless, the model soups of SSE/StarSSE remain effective even on OOD data (see Figure 4).
>
> [1] Fort et. al. Deep ensembles: A loss landscape perspective. 2019.
>
> \
> We hope we adequately addressed all your concerns and that you would consider reflecting that in your score. If you have any additional questions, we would be happy to answer them during the discussion period =)

---

> > ### Comment · Reviewer_XbMH · 2023-08-14
> > **Thank you.**
> >
> > Thank you for your rebuttal. I am happy with the work done to address the weakensses that I noted, and I am now convinced that StarSSE provides a meaningful improvement to ensemble performance in the transfer learning setting. Likewise, I am happy with the way the authors have addresed my questions, and I find the results concerning ensemble diversity to be quite interesting in their own right, in particularly the fact that SSE/StarSSE produce ensembles that are more diverse than LocalDE. I look forwards to seeing an updated version of this paper with these results (and those discussed in your response to Reviewer rBpj), and I have updated my score.

---

### Author Rebuttal · Authors · 2023-08-09

First of all, we want to thank all the reviewers for their constructive and valuable feedback! We have found all the suggestions very relevant and will definitely incorporate them into the paper.

We are happy that our analysis of the effectiveness of local and semi-local ensemble methods in the transfer learning setup and the resulting insights on how staying or leaving the pre-train basin influences the ensemble quality were found convincing, interesting, and important. We see this analysis and insights as the main contributions of our paper since they show which loss landscape exploration tactics are suitable for the transfer learning setup and can be useful in practice: while local ensemble methods can be beneficial, existing semi-local methods can not leave the pre-train basin without losing the transfer learning benefits.

Most of the reviewers' comments concerned the effectiveness of StarSSE, the proposed modification of the SSE method for transfer learning. We did not closely focus on this part in the submission because we looked at StarSSE more as an example of local ensemble methods that helps us to show that local methods, if used appropriately, can be beneficial in transfer learning too. However, we agree that some additional experiments and comparisons would improve this part of the paper. Given the tight rebuttal period, we do our best to address most of the concerns and provide additional results in the replies below. We will include more detailed additional experiments in the final revision of the paper.

---

### Decision · Program_Chairs · 2023-09-21

**Decision:**

Accept (poster)

**Comment:**

This paper studies improving an ensemble obtained from finetuning a single pretrained model. It is commonly believed an ensemble obtained from a single pretrained model is restricted to the same basin in the loss landscape and, thus, is not useful. Recent work on SnapShot Ensembling (SSE) suggests training an ensemble with a cyclic learning rate to help the original model escape the current basin. Although this semi-global approach can achieve better diversity compared to purely local methods, the authors show that, in the transfer learning setup, SSE could make the ensemble quality degrade. The ensemble degrades because ensemble members keep moving away from the pretrain basin, especially when a higher learning rate or a larger ensemble size is used. Thus, The paper proposes StarSSE, a parallel version of the original SSE method. It is shown that by converting the SSE from sequential to parallel, the new method effectively achieves better diversity without forcing the ensemble members to move too far from the pretrain basin.

The authors also provide a good rebuttal and improve the paper by providing a few extra experiments: (1) More clear quantitative comparison of SSE and StarSSE ensembles. (2) Showing StarSSE achieves better diversity than local deep ensemble methods. (3) Comparing StarSSE to other local ensembling methods regarding test accuracy. (4) StarSSE on more downstream tasks. (5) StarSSE and baselines with hyperparameter and augmentation diversification.